# WEAK-TO-STRONG PREFERENCE OPTIMIZATION: STEALING REWARD FROM WEAK ALIGNED MODEL

**Wenhong Zhu**[1,2]**, Zhiwei He**[1]**, Xiaofeng Wang**[1]**, Pengfei Liu**[1,2]**, Rui Wang**[1]*

[1]Shanghai Jiao Tong University, [2]Shanghai Innovation Institute
{zwhong714, wangrui12}@sjtu.edu.cn

## ABSTRACT

Aligning language models (LMs) with human preferences has become a key area of research, enabling these models to meet diverse user needs better. Inspired by weak-to-strong generalization, where a strong LM fine-tuned on labels generated by a weaker model can consistently outperform its weak supervisor, we extend this idea to model alignment. In this work, we observe that the alignment behavior in weaker models can be effectively transferred to stronger models and even exhibit an amplification effect. Based on this insight, we propose a method called *Weak-to-Strong Preference Optimization (WSPO)*, which achieves strong model alignment by learning the distribution differences before and after the alignment of the weak model. Experiments demonstrate that WSPO delivers outstanding performance, improving the win rate of Qwen2-7B-**Instruct** on Arena-Hard from 39.70 to **49.60**, achieving a remarkable **47.04** length-controlled win rate on AlpacaEval 2, and scoring **7.33** on MT-bench. Our results suggest that using the weak model to elicit a strong model with a high alignment ability is feasible. The code is available at https://github.com/zwhong714/weak-to-strong-preference-optimization.

## 1 INTRODUCTION

Cutting-edge large language models (LLMs) are trained through a three-phase process (OpenAI, 2024). Initially, these models undergo pre-training on extensive corpora, using next-token prediction to build a foundational understanding (Radford et al., 2018; 2019). Following this, the pre-trained models are fine-tuned using supervised fine-tuning (SFT) to better align with specific instructions (Wei et al., 2021). However, these models have flaws, as they can sometimes produce factual inaccuracies, exhibit biases, and display other undesirable behaviors (Bai et al., 2022; Liu et al., 2024b). Learning from human preferences (Christiano et al., 2017) is a paradigm in the final phase aiming to better align pre-trained and instruction-followed generative LMs with human values and goals.

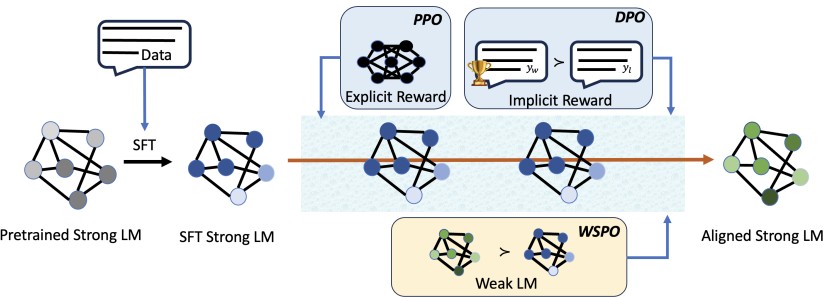

Figure 1: Pipeline for LM alignment. (1) Perform SFT on the pre-trained model using expert data. (2) Current approaches incorporate explicit or implicit reward mechanisms to fine-tune the model further, aligning its behavior with human preferences. (3) WSPO aligns strong models by utilizing the distributional differences observed before and after aligning the weak model.

---

*Corresponding author.

As shown in Figure 1, the alignment method in RLHF traditionally involves training an explicit scalar-valued reward model that captures human judgment. This reward model is then used to fine-tune the LM through reinforcement learning (RL) (Christiano et al., 2017), such as proximal policy optimization (PPO) (Schulman et al., 2017) algorithm. This pipeline is considerably more complex than SFT, involving training multiple LMs and sampling from the LM policy in the training loop, incurring significant computational costs. More recent research has explored alignment approaches that eliminate the need for a separate reward model, instead aligning the LM directly based on human preferences, named Direct Preference Optimization (DPO) (Rafailov et al., 2024).

Learning from human feedback preferences, whether online or offline, is crucial in PPO and DPO. A phenomenon known as *weak-to-strong generalization* (Burns et al., 2023) demonstrates that a strongly pre-trained model, when fine-tuned on labels generated by a weaker model, consistently outperforms the weaker supervised model. This intriguing result prompts the question: *Can we leverage the alignment signal from the weak models to align a strong model?*

This paper introduces a novel method called *Weak-to-Strong Preference Optimization (WSPO)*, a loss function designed to effectively transfer the alignment capability from a weaker model to a stronger one. Our results show that the stronger model can amplify this transferred alignment. Instead of using data generated by the weaker model as labels for aligning the stronger model, we establish a relationship between the weak model (serving as a reward model) and the strong model in the context of RL optimization. By learning the differences before and after the alignment of the weak model, we can effectively enhance the alignment ability of the stronger model.

The main contributions of this paper are summarized as follows:

- We introduce the WSPO method, a loss function that transfers the alignment capability of the weak model to the strong model by learning the distributional differences before and after the weak model's alignment.
- We find that the alignment capability of the weaker model can be effectively transferred to the stronger model, amplifying the stronger model's alignment performance.
- Our experimental analysis reveals that the proposed method improves the win rate of Qwen2-7B-**Instruct** on Arena-Hard from 39.70 to **49.60**, achieving an impressive **47.04** length-controlled win rate on AlpacaEval 2, and obtaining a score of **7.33** on MT-bench. Results on various common sense, mathematical, and other reasoning tasks demonstrate that our method preserves the knowledge embedded in the strong model.

## 2 PRELIMINARIES

Given a query sequence $x := (x_1, \ldots, x_m) \in \mathcal{X}$, an auto-regressive LM defines a probability distribution over possible response sequences $y := (y_1, y_2, \ldots, y_n) \in \mathcal{Y}$. The probability $\pi_\theta(y \mid x)$ can be decomposed using the chain rule of probability as $\pi_\theta(y \mid x) = \prod_{t=1}^{n} \pi_\theta(y_t \mid y_{<t}, x)$, where $y_{<t}$ denotes $\{y_1, y_2, ..., y_{t-1}\}$. Typically, an LM is pre-trained on a large, unlabeled text dataset using maximum likelihood estimation (MLE). This process can be viewed as learning a distribution that narrows the gap with the true data distribution.

### 2.1 SUPERVISED FINE-TUNING

Following initialization with a pre-trained LM $\pi_{\text{base}}$, the model undergoes further fine-tuning on smaller, meticulously curated datasets containing expert demonstrations of high-quality responses. This results in the model $\pi_{\text{sft}}$. These datasets emphasize desired behaviors such as following instructions (Wei et al., 2021), engaging in dialogue (Li et al., 2016), and other similar tasks.

### 2.2 FINETUNING FROM HUMAN FEEDBACK

Learning from human feedback (Christiano et al., 2017) has garnered significant attention due to its potential to use human-labeled datasets for aligning LMs with human preferences (Wei et al., 2021). In these alignment approaches, the optimization objective is generally to maximize the expected reward from an implicit or explicit reward function while including a KL-divergence term from the reference policy as a penalty for divergence (Shi et al., 2024).

### 2.2.1 RLHF

**Learning the reward model.** Learning a reward model typically involves training a binary classifier to distinguish between preferred and less preferred actions using a logistic regression loss. A commonly used classifier for this task is the Bradley-Terry model (David, 1963). In this model, for a given context $x$ and response $y$, the pointwise reward of selecting $y$ given $x$ is denoted by $r(x, y)$.

**Policy Optimization with the learned reward.** Once the reward model is established, the model alignment process maximizes the expected reward while preserving the original distribution $\pi_{\text{ref}}$. This is often achieved using a family of $f$-divergence regularization methods (Rafailov et al., 2024; Shi et al., 2024). For example, when using KL divergence, the optimization problem on a static dataset of comparisons $\mathcal{D} = \left\{ x^{(i)}, y_w^{(i)}, y_l^{(i)} \right\}_{i=1}^{N}$, where $y_w$ and $y_l$ represent the preferred and dispreferred completions, respectively, can be formulated as follows:

$$\max_{\pi_\theta} \mathbb{E}_{x \sim \mathcal{D}, y \sim \pi_\theta(y|x)} \left[ r(x, y) \right] - \beta \mathbb{D}_{\text{KL}} \left[ \pi_\theta(y \mid x) \| \pi_{\text{ref}}(y \mid x) \right], \tag{1}$$

where $\beta$ is a parameter that controls the degree of deviation from the reference policy $\pi_{\text{ref}}$. If $\beta$ is set too high, KL regularization forces the aligned model to closely mimic the SFT model, potentially limiting the effectiveness of the alignment (Geist et al., 2019). On the other hand, if $\beta$ is set too low, the aligned model may diverge excessively from the SFT model, leading to reward hacking (Skalse et al., 2022). This overfitting problem can compromise critical capabilities developed during pretraining or SFT (Stiennon et al., 2020).

### 2.2.2 DPO

An alternative approach to learning from the human preference paradigm described above is DPO (Rafailov et al., 2024), which completely bypasses the need to train a reward model. The loss function that DPO optimizes is expressed as a function of $\pi_\theta$ as follows:

$$\mathcal{L}_{\text{DPO}} \left( \pi_\theta; \pi_{\text{ref}} \right) = -\mathbb{E}_{(x, y_w, y_l) \sim \mathcal{D}} \left[ \log \sigma \left( \beta \log \frac{\pi_\theta \left( y_w \mid x \right)}{\pi_{\text{ref}} \left( y_w \mid x \right)} - \beta \log \frac{\pi_\theta \left( y_l \mid x \right)}{\pi_{\text{ref}} \left( y_l \mid x \right)} \right) \right]. \tag{2}$$

While DPO simplifies the process by bypassing the need for reward function training, this may result in a final strategy that is less regularized and robust compared to RLHF. In RLHF, the underfitted reward function is crucial in balancing and optimizing the final strategy (Azar et al., 2024).

## 3 METHOD

Building on the concept of *weak-to-strong generalization*, where a strong model is capable of generalizing beyond weak labels rather than simply imitating the behavior of the weaker model, in this section, we leverage RL theory to demonstrate that it is possible to train a strong alignment model using the distributional differences before and after weak model alignment.

### 3.1 YOUR LANGUAGE MODEL IS SECRETLY A REWARD MODEL

Prior work (Rafailov et al., 2024) shows that given a specific reward model $r(x, y)$, the optimal solution to the KL-constrained reward maximization problem in Objective 1 takes the form:

$$\pi_r(y \mid x) = \frac{1}{Z(x)} \pi_{\text{ref}}(y \mid x) \exp \left( \frac{1}{\beta} r(x, y) \right), \tag{3}$$

where $Z(x) = \sum_y \pi_{\text{ref}}(y \mid x) \exp \left( \frac{1}{\beta} r(x, y) \right)$ is the partition function, and $\pi_r(y \mid x)$ represents the model after alignment. The reward function can be shown as follows by rearranging Equation 3:

$$r(x, y) = \beta \log \frac{\pi_r(y \mid x)}{\pi_{\text{ref}}(y \mid x)} + \beta \log Z(x). \tag{4}$$

**Theorem 1.** *Under mild assumptions, all reward classes consistent with the Plackett-Luce (and Bradley-Terry in particular) models can be represented with the reparameterization $r(x, y) = \beta \log \frac{\pi(y|x)}{\pi_{ref}(y|x)}$ for some model $\pi(y \mid x)$ and a given reference model $\pi_{ref}(y \mid x)$.*

Based on Theorem 1 proposed by Rafailov et al. (2024), we know that the reward function can be expressed as the difference in distributions before and after model alignment.

## 3.2 WEAK-TO-STRONG PREFERENCE OPTIMIZATION

Before introducing our method, we define $\pi_r^{\text{weak}}(y \mid x)$ as a weak model aligned using specific algorithms, such as PPO or DPO. Similarly, $\pi_{\text{ref}}^{\text{weak}}(y \mid x)$ denotes the reference model, which may correspond to either $\pi_{\text{sft}}$ or $\pi_{\text{base}}$. These notations also apply to strong models.

**Derive the WSPO objective.** Theorem 1 demonstrates that a reward model trained on the preference dataset $\mathcal{D}$ can be expressed as the distribution difference before and after model alignment. Consequently, we can align a weak model and derive an aligned weak model $\pi_r^{\text{weak}}(y \mid x)$. Thus, the reward model $r(x, y)$ can be formulated as $\beta \log \frac{\pi_r^{\text{weak}}(y|x)}{\pi_{\text{ref}}^{\text{weak}}(y|x)}$. Next, we employ this transformed reward model to align a strong model $\pi_{\text{ref}}^{\text{strong}}(y \mid x)$, allowing us to derive that

$$\pi_r^{\text{strong}}(y \mid x) = \frac{1}{Z'(x)} \pi_{\text{ref}}^{\text{strong}}(y \mid x) \exp\left(\frac{1}{\lambda} r(x, y)\right) \propto \pi_{\text{ref}}^{\text{strong}}(y \mid x) \left(\frac{\pi_r^{\text{weak}}(y \mid x)}{\pi_{\text{ref}}^{\text{weak}}(y \mid x)}\right)^{1/\gamma}, \quad (5)$$

where $Z'(x) = \sum_y \pi_{\text{ref}}^{\text{strong}}(y \mid x) \exp\left(\frac{1}{\lambda} r(x, y)\right)$, $\lambda$ is the regularization strength to align strong LM using Obejctive 1, and $\gamma$ equals to $\lambda/\beta$. Although $r(x, y)$ is analytically tractable, substituting it into Obejctive 1 for policy optimization poses challenges due to the absence of a partition function. However, we can optimize Equation 5 to minimize the difference in distance between the logarithmic probability distributions before and after aligning the strong and weak models. Therefore we obtain

$$\mathcal{L}_{\text{WSPO}} = \mathbb{E}_{(x,y)\sim\mathcal{D}} \left[\frac{1}{|y|} \left\|\gamma \log \frac{\pi_\theta^{\text{strong}}(y \mid x)}{\pi_{\text{ref}}^{\text{strong}}(y \mid x)} - \log \frac{\pi_r^{\text{weak}}(y \mid x)}{\pi_{\text{ref}}^{\text{weak}}(y \mid x)}\right\|_2^2\right]. \quad (6)$$

Derivation see in Appendix A.3. An intuitive explanation is that we leverage the change in the weak model's alignment before and after as a supervisory signal to guide the alignment of the stronger reference model.

**The role of the hyperparameter $\gamma$.** The hyperparameter $\gamma$ plays a dual role: it maximizes the reward function $\beta \log \frac{\pi_r^{\text{weak}}(y|x)}{\pi_{\text{ref}}^{\text{weak}}(y|x)}$, while simultaneously constraining the proximity of the original distribution $\pi_{\text{ref}}^{\text{strong}}(y \mid x)$ to the optimized distribution $\pi_\theta^{\text{strong}}(y \mid x)$.

**What does the WSPO do?** The gradient with respect to the parameters $\theta$ can be written as:

$$\mathbb{E}_{(x,y)\sim\mathcal{D}} \sum_{t=1}^{|y|} \left[\frac{2}{|y|} \left(\gamma \log \frac{\pi_\theta^{\text{strong}}(y_{<t} \mid x)}{\pi_{\text{ref}}^{\text{strong}}(y_{<t} \mid x)} - \log \frac{\pi_r^{\text{weak}}(y_{<t} \mid x)}{\pi_{\text{ref}}^{\text{weak}}(y_{<t} \mid x)}\right) \nabla_\theta \log \pi_\theta^{\text{strong}}(y_{<t} \mid x)\right]. \quad (7)$$

As shown in Equation 7, the direction of the gradient is influenced by the contents of the parentheses, while the extent of gradient descent is not dictated by the model's likelihood on the dataset. Derivation see in Appendix A.4. This may help mitigate the overfitting problem commonly associated with DPO alignment algorithms (Azar et al., 2024).

**WSPO outline.** The general WSPO pipeline operates as follows: 1.) Utilize offline datasets $\mathcal{D} = \left\{x^{(i)}, y_w^{(i)}\right\}_{i=1}^{N}$, such as the selected preference or SFT datasets; paired datasets are not required.

In Appendix C, we demonstrate that even the rejected preference dataset remains effective for the WSPO algorithm. 2.) Prepare the weak model, both pre-and post-alignment. 3.) Optimize the LM $\pi_\theta^{\text{strong}}(y \mid x)$ to minimize the objective $\mathcal{L}_{\text{WSPO}}$ for the specified dataset. The only parameter requiring tuning is $\gamma$.

## 4 EXPERIMENTS

This section empirically evaluates WSPO's ability to align strong models by learning from weaker ones. Our findings reveal that aligned weak models can successfully transfer their alignment behaviors to stronger models, often resulting in an enhanced alignment effect. Additionally, WSPO demonstrates competitive performance compared to strong models trained using PPO and DPO. We begin by illustrating the feasibility of our method with a toy example. Next, we analyze the stability of WSPO algorithm training. Finally, we conduct a comprehensive evaluation of the algorithm's overall performance.

### 4.1 SUMMARIZATION WITH A LENGTH REWARD

#### 4.1.1 EXPERIMENT SETUP

**Task.** We employ a toy summarization task with a hardcoded reward function that incentivizes models to generate summaries with lengths falling within the range of $[L_{\min}, L_{\max}]$:

$$r(x, y) := \begin{cases} 0, & \text{if } |y| \in [L_{\min}, L_{\max}] \\ -1, & \text{otherwise} \end{cases} \tag{8}$$

**Models.** We employ the Qwen2-1.5B-base and Qwen2-7B-base models (Yang et al., 2024a) as our pre-trained weak model, $\pi_{\text{base}}^{\text{weak}}$, and strong model, $\pi_{\text{base}}^{\text{strong}}$, respectively. To train the SFT models, $\pi_{\text{sft}}^{\text{weak}}$ and $\pi_{\text{sft}}^{\text{strong}}$, we utilize the training split of the XSUM dataset (Narayan et al., 2018). Subsequently, the validation split is employed for further fine-tuning, leading to the development of the corresponding PPO-aligned models, $\pi_{\text{ppo}}^{\text{weak}}$ and $\pi_{\text{ppo}}^{\text{strong}}$. When training with WSPO, we directly use the distributional differences between $\pi_{\text{base}}^{\text{weak}}$ and $\pi_{\text{ppo}}^{\text{weak}}$ to align $\pi_{\text{base}}^{\text{strong}}$ and derive $\pi_{\text{wspo}}^{\text{strong}}$, because no additional knowledge is required to output a summary in the summary task.

**Evaluation.** The parameters $L_{\min}$ and $L_{\max}$ are set to 20 and 30, respectively. We use the test split for evaluation to guarantee no data contamination (Zhu et al., 2023). Detailed experimental settings are provided in Appendix B.1.

#### 4.1.2 RESULTS AND ANALYSIS

We visualize the lengths generated by various alignment algorithms across models of different sizes and calculate the win rate, which represents the proportion of lengths falling within the $[L_{\min}, L_{\max}]$ range.

**WSPO exhibits comparable alignment capability against PPO.** As shown in Figure 2, the PPO and WSPO alignment algorithms address variable generation lengths arising from SFT training. The left figure illustrates how our method, using the greedy decoding algorithm, performs similarly to the PPO algorithm and can effectively mitigate outliers. In the right figure, we present the averaged results from three samplings. From the error lines, it is evident that both PPO and WSPO exhibit high stability during generation. Notably, our method is better at lower temperatures than the performance of the PPO-aligned strong model.

**WSPO embraces certain generalizations.** On the left in Figure 2, we can observe that the weak model performs quite well on the alignment task, with the results predominantly concentrated in the reward area. Additionally, we can see that the strong LM, obtained through WSPO alignment with the weak model, exhibits a different length distribution in its generated outputs. This indicates that the strong LM is not merely imitating the behavior of weak models.

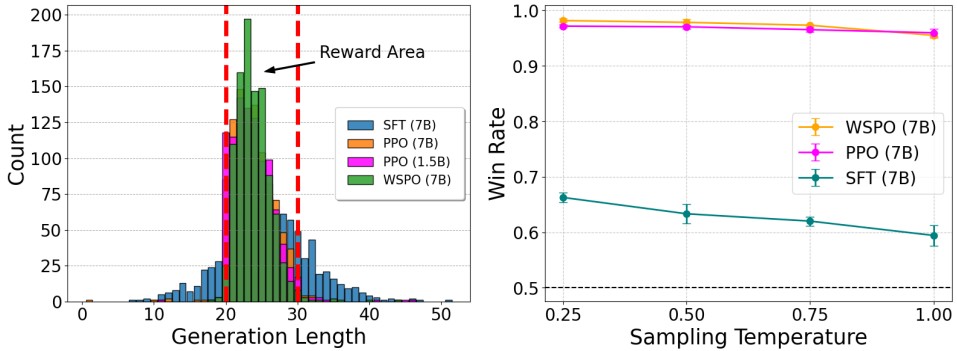

Figure 2: **Left.** PPO and WSPO alignment methods vary in the length of generated sequences compared to the reference SFT model using greedy decoding. **Right.** PPO and WSPO alignment methods show variation in reward hits compared to the reference SFT model, using the top-$p$ sampling algorithm at different temperatures.

**WSPO provides a faster alignment process.** Traditionally, model alignment involves performing SFT on pre-trained LMs first, then applying PPO algorithms for alignment. In this section, we merge these two stages. Using the same amount of alignment data typically reserved for PPO training, WSPO achieves comparable alignment effects in a more streamlined process. This demonstrates that a pre-trained model can effectively perform alignment by learning differential distribution signals, provided it has acquired sufficient knowledge.

## 4.2 SINGLE-TURN DIALOGUE

### 4.2.1 EXPERIMENTAL SETUP

**Task.** In single-turn dialogue, when prompt $x$ is a human query, the model must either politely respond to the query or refuse to answer.

**Models.** In this scenario, we fine-tune the Qwen2-1.5B-base and Qwen2-7B-base models (Yang et al., 2024a) exclusively on the preferred completions, resulting in two models: $\pi_{\text{sft}}^{\text{weak}}$ and $\pi_{\text{sft}}^{\text{strong}}$. We refer to these models collectively as **Preferred-FT** models. Subsequently, we apply the DPO algorithm to perform alignment training using the Anthropic Helpful and Harmless conversation training dataset (Bai et al., 2022). This step produces two alignment models: $\pi_{\text{dpo}}^{\text{weak}}$ and $\pi_{\text{dpo}}^{\text{strong}}$. To address potential biases introduced by the base models pretrained on the high-quality corpus, we utilize the distributional differences between $\pi_{\text{sft}}^{\text{weak}}$ and $\pi_{\text{dpo}}^{\text{weak}}$. These differences guide the alignment of $\pi_{\text{sft}}^{\text{strong}}$ through the WSPO algorithm, ultimately resulting in the model $\pi_{\text{wspo}}^{\text{strong}}$.

**Evaluation.** We use the test split of the Anthropic HH dataset to assess alignment performance, evaluating through a single-step human-assistant interaction. The evaluation leverages preferred completions from the test set as references to calculate the win rates of different methods. Taking cost into consideration, we selected the GPT-4o-mini as the judge model. We use Qwen2.5-72B (Qwen, 2024) to verify the validity of the evaluation. Detailed experimental settings are provided in Appendix B.2.

### 4.2.2 RESULTS AND ANALYSIS

**The alignment signal from the weak model plays a crucial role.** As shown on the left side of Figure 3, unlike DPO, which requires distinguishing the likelihood between preferred and dispreferred pairs in preference data, WSPO, based on the **Preferred-FT** model, continues to leverage the alignment signal from the weak model on **preferred data** to perform alignment. Consequently, WSPO adjusts the distribution learned exclusively from the preferred data, resulting in improved alignment ability compared to the Preferred-FT model.

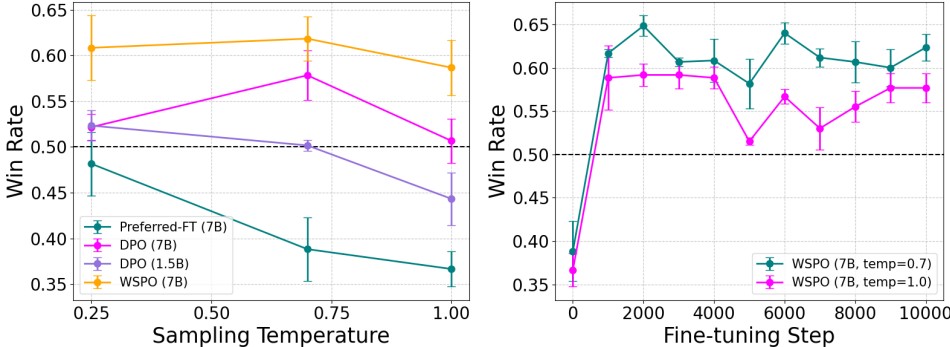

Figure 3: **Left.** Win rates computed by GPT-4o-mini for Anthropic-HH single-step dialogue at different temperatures. **Right.** The win rates for different sampling temperatures remain relatively stable throughout the training process. WSPO demonstrates consistent performance across varying sampling temperatures over time.

**WSPO exhibits better alignment capability compared to DPO.** Tuning the parameters for DPO proves to be quite challenging, as seen in Appendix B.2. After several adjustments, we achieved a relatively favorable outcome with the hyperparameter $\beta = 0.5$, which was determined by testing a range of values $\{0.1, 0.5, 1.0, 2.0, 5.0\}$ for the 7B model. As illustrated on the left side of Figure 3, surprisingly, our model demonstrates better alignment performance than DPO with the hyperparameter $\gamma = 0.1$.

**WSPO exhibits fast convergence and stability.** As shown in the right part of Figure 3, the model achieves a high win rate after just 1k fine-tuning steps, demonstrating fast convergence. Additionally, the training process remains stable throughout.

### 4.3 A COMPLEX EVALUATION

#### 4.3.1 EXPERIMENTAL SETUP

**Models.** We train the **base models** on the UltraChat-200k (Ding et al., 2023) dataset to obtain the SFT models. We use the off-the-shelf **instruction-tuned** models as the SFT models. Then, we perform alignment using DPO on the UltraFeedback dataset (Cui et al., 2024) using the SFT model as the starting point. The process of obtaining the models is largely consistent with that of Exp 4.2.

**Evaluation.** We evaluate our models primarily on three widely-adopted, open-ended instruction-following benchmarks: MT-Bench (Zheng et al., 2024a), AlpacaEval 2 (Dubois et al., 2024), and Arena-Hard v0.1 (Li et al., 2024c). These benchmarks assess the models' conversational versatility across diverse queries and are commonly used by the community (Meng et al., 2024). Besides, fine-tuning LMs is challenging, notably since it can cause forgetting (French, 1992) of pre-trained knowledge. To demonstrate that the strong model can generalize beyond weak labels instead of merely imitating the behavior of weak models. We use zero-shot or few-shot learning to test the reasoning ability across five benchmarks, including MMLU (Hendrycks et al., 2021), CMMLU (Li et al., 2024a), Truthful-QA (Lin et al., 2021), GSM-PLUS (Li et al., 2024b), and GSM8K (Cobbe et al., 2021). We evaluate these benchmarks by using *llm-evaluation-harness* (Gao et al., 2024) repo. Evaluation details are in Appendix B.3.

#### 4.3.2 RESULTS AND ANALYSIS

As shown in Table 1, the Instruct setting consistently outperforms the Base setting. This is primarily because the Instruct model utilizes high-quality demonstration and preference data for SFT and RLHF (Yang et al., 2024a).

**DPO shows limited improvement for the Instruct model.** The SFT-trained model demonstrates limited effectiveness across the three benchmarks, highlighting areas for potential enhancement.

Table 1: Evaluation results of models across different settings and benchmarks. LC and WR refer to length-controlled and raw win rates, respectively. We train SFT models under the Base settings using the UltraChat-200K dataset. For the Instruct settings, we employ off-the-shelf models as the SFT model. The SFT and DPO versions of the weak model are employed to align the strong model within the WSPO algorithm.

| Method | Qwen2-Base (1.5B) | | | | Qwen2-Instruct (1.5B) | | | |
|---|---|---|---|---|---|---|---|---|
| | AlpacaEval2 | | Arena-Hard | MT-Bench | AlpacaEval2 | | Arena-Hard | MT-Bench |
| | LC (%) | WR (%) | WR (%) | Score | LC (%) | WR (%) | WR (%) | Score |
| SFT | 4.16 | 2.30 | 0.90 | 4.68 | 5.31 | 3.42 | 2.40 | 5.05 |
| DPO | 5.56 | 4.79 | 2.60 | 5.03 | 8.93 | 6.77 | 4.00 | 5.60 |
| **Method** | **Qwen2-Base (7B)** | | | | **Qwen2-Instruct (7B)** | | | |
| | AlpacaEval2 | | Arena-Hard | MT-Bench | AlpacaEval2 | | Arena-Hard | MT-Bench |
| | LC (%) | WR (%) | WR (%) | Score | LC (%) | WR (%) | WR (%) | Score |
| SFT | 11.54 | 5.65 | 5.30 | 5.86 | 30.73 | 28.32 | 39.70 | 7.19 |
| DPO | 14.06 | 8.45 | 10.70 | 6.70 | 32.10 | 28.15 | 39.30 | 7.26 |
| WSPO | **26.77** | **26.68** | **29.00** | **7.00** | **47.04** | **48.32** | **49.60** | **7.33** |

While DPO provides noticeable improvements for the base and weaker models, performance declines with Qwen2-7B-Instruct. This decline could be due to Qwen2-7B's ability to achieve strong alignment through raw, high-quality data and complex RLHF processes (Yang et al., 2024a). Relying solely on the Ultrafeedback dataset for DPO learning might not lead to performance gains, as the dataset may already be part of its original high-quality data. Additionally, it's possible that DPO adversely affected the model's initial performance.

**WSPO effectively learns and amplifies the alignment signals of weak models.** With WSPO, the strong model consistently delivers great results across all three benchmarks. The impressive performance of WSPO can be attributed to its unique approach: unlike DPO, which learns directly from preference data pairs, WSPO derives alignment signals from weaker models, not only dependent on the Ultrafeedback dataset itself. For instance, on the Qwen2-1.5B-Instruct, the alignment ability of weak models improved from 2.40 to 4.00 with DPO learning, as measured by the Arena-Hard evaluation. Subsequently, the strong model's alignment capability was amplified from 39.70 to **49.60** by leveraging the differences in alignment signals from the weak model—something that DPO learning on datasets alone cannot achieve. The amplification phenomenon might be attributed to the limited parameter size of the weak model, which constrains its ability to achieve optimal alignment. However, transferring this alignment to stronger models could offer substantial benefits. Additionally, our method circumvents direct training on the preference dataset, effectively reducing risks such as overfitting and reward hacking.

Table 2: Evaluation results of models across different benchmarks. We evaluate these benchmarks by using *llm-evaluation-harness* (Gao et al., 2024) repo.

| Model | MMLU | CMMLU | Truthful-QA | GSM-PLUS | GSM8K | Avg. |
|---|---|---|---|---|---|---|
| Qwen2-1.5B-Instruct | 55.70 | 69.62 | 28.52 | 38.83 | 59.78 | 50.49 |
| Qwen2-7B-Base | 69.43 | **83.34** | 37.33 | 57.39 | 79.83 | 65.46 |
| Qwen2-7B-Instruct | **69.94** | 81.84 | 41.00 | 56.91 | 77.86 | 65.51 |
| Qwen2-7B-Instruct + WSPO | 69.44 | 80.82 | **47.00** | 57.96 | 77.94 | 66.63 |
| Qwen2-7B-Base + WSPO | 69.37 | 80.98 | 44.68 | **60.06** | **81.31** | **67.28** |

**WSPO generalizes beyond weak models rather than simply imitating them.** As shown in Table 2, Qwen2-1.5B-Instruct is much less capable than the 7B version, further demonstrating that WSPO prevents knowledge forgetting in common sense, mathematics and other reasoning tasks while enhancing the model's overall alignment ability, as shown in Table 1. Notably, on the **TruthfulQA** dataset, both the base model and the Instruct model exhibited improved capabilities in assessing the degree of truthfulness.

# 5 ANALYSIS

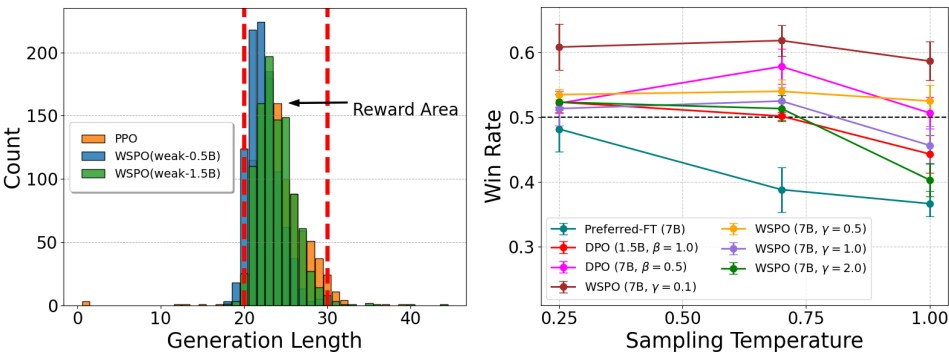

Figure 4: **Left.** The effect of weak model size on the sequence length generated by WSPO compared to the PPO using greedy decoding. **Right.** The impact of different $\gamma$ hyperparameters on WSPO in a single-turn dialogue analysis.

## 5.1 IMPACT OF WEAK MODEL

As discussed in the previous section, we utilized the probability difference between a weak base model and its aligned version to align a stronger model. In this section, we empirically investigate the impact of model size by using the Qwen2-0.5B model as a weaker counterpart to the Qwen2-1.5B model, aiming to explore how model size affects alignment strength. The experimental setup mirrors that of Section 4.1.1. As illustrated on the left side of Figure 4, even a weaker model can provide a robust alignment signal to a stronger model (e.g., the Qwen2-7B model). Furthermore, in the Instruct setting of Section 4.3, we use the 0.5B model as the weaker model without any alignment enhancement following DPO training. When this alignment is transferred to a stronger model, it achieves a score of **45.00** on the Arena-hard benchmark using WSPO optimization. This indicates that parameter size may limit alignment in weaker models, whereas stronger models can amplify this alignment. Besides, the alignment ability of a weak model is also important, we can have fine-grained alignment on a weak model and then migrate the alignment ability to a strong model to achieve better alignment.

## 5.2 IMPACT OF HYPERPARAMETER

Recalling the WSPO loss in Equation 6, we introduce the hyperparameter $\gamma$, which represents the ratio of regularization intensity applied to the strong and weak models in the optimization objective outlined in Objective 1, as well as the penalty for deviating from the original distribution. This section investigates the impact of $\gamma$ on the alignment strength. We test values of $\gamma \in \{0.1, 0.5, 1.0, 2.0\}$ to evaluate its effect on regularization. As illustrated on the right side of Figure 4, adjusting $\gamma$ enables us to modulate the degree to which the stronger model aligns with the weaker one and deviates from the original distribution. When $\gamma = 1$, the alignment of the strong model closely mirrors that of the weak model. As $\gamma$ increases beyond 1, the strong model's alignment increasingly favors the original distribution. Conversely, the strong models exhibit superior alignment when $\gamma$ is less than 1. Therefore, despite $\gamma$ incorporating a penalty for deviations from the original distribution, we can infer that the strong model requires a smaller regularization than the weak model when optimizing the objective function in Objective 1.

## 5.3 IMPACT OF SFT PHASE

We also leveraged the probability difference between Qwen2-1.5-Base and Qwen2-1.5-Instruct to align stronger models from the Base version directly. On the Arena-Hard benchmark, the Base model initially scored 7.70. However, after applying the WSPO algorithm for alignment with Ultrafeedback, the score saw a modest improvement to 9.30. This limited gain underscores the significance of high-quality knowledge injection during the SFT phase.

# 6 RELATED WORK

## 6.1 TRAINING-TIME ALIGNMENT

RLHF is a technique designed to align LLMs with human preferences and values (Christiano et al., 2017; Bai et al., 2022). In the third stage of RLHF, the PPO algorithm (Schulman et al., 2017) is commonly used. Recent advancements, such as Reinforcement Learning with AI Feedback (RLAIF), offer potential alternatives to traditional human feedback methods (Pang et al., 2023). However, challenges throughout the RLHF pipeline, from preference data collection to model training, have been noted by Radford et al. (2018). In contrast, approaches like DPO (Rafailov et al., 2024) bypass the need for a reward model by directly training LLMs using human preferences. Other competing methods, such as IPO (Azar et al., 2024), KTO (Ethayarajh et al.), and ORPO (Hong et al., 2024), have also emerged.

## 6.2 INFERENCE-TIME ALIGNMENT

Decoding strategies aim to generate text continuations that balance diversity and coherence (Zhu et al., 2024). Some methods trade off computational efficiency during inference to better align with human preferences. The simplest of these is the Best-of-$N$ approach, which involves sampling multiple outputs from $\pi_{\text{ref}}$ and selecting the one with the highest reward according to a reward model (Touvron et al., 2023). Another approach is Emulated Fine-Tuning (EFT) (Mitchell et al., 2023), a scale-decoupling method that transfers fine-tuning effects between small and large LMs. Liu et al. (2024a) demonstrated the empirical effectiveness of this proxy-tuning technique, showing it rivals standard fine-tuning across various benchmarks. Additionally, Liu et al. (2024b) introduced DeRa, a cost-efficient method that dynamically adjusts alignment strength during inference. Zhou et al. (2024) used the log-probability difference between small-tuned and untuned models to guide a frozen large model, providing an efficient up-scaling strategy without fine-tuning.

## 6.3 WEAK-TO-STRONG GENERALIZATION

Several works have been proposed to use weak model supervision to elicit the capabilities of a much stronger model. Burns et al. (2023) found that strong models fine-tuned by weak supervisors consistently outperform their weak counterparts. Yang et al. (2024b) presents a method that improves model reasoning by employing weak supervision to autonomously refine training data autonomously, enabling the expansion of reasoning abilities without human annotations or advanced models. Unlike these approaches, our method uses weak model supervision for alignment to enhance helpfulness while maintaining the strong model's original ability.

# 7 DISCUSSION

**Conclusion.** This paper introduced WSPO, a method for transferring alignment capabilities from a weaker model to a stronger one by leveraging distributional differences before and after weak model alignment. Experimental results show that WSPO improves model performance on key benchmarks, offering an efficient alternative to traditional alignment methods.

**Limitations and future work.** We did not explore the alignment transfer properties across different language model architectures or examine the impact of weak model alignment strength in WSPO. Our study also does not explain why transferring a weak model's alignment ability to a stronger model amplifies it. Future research could investigate the use of weak models as reward models in reinforcement learning frameworks to facilitate alignment or seek to explain this phenomenon.

## ACKNOWLEDGMENTS

This paper is supported by the General Program of National Natural Science Foundation of China (62176153).

ETHICS STATEMENT

Although the datasets used in this paper are open-source and helpful, we did not perform an in-depth evaluation of them, nor did we account for factors such as safety, honesty, or other considerations when designing the WSPO loss function.

REPRODUCIBILITY STATEMENT

All training experiments in this paper were conducted using 8×H100 GPUs, leveraging the LLaMA-Factory (Zheng et al., 2024b) repository, which offers an integrated framework for fine-tuning over 100 LLMs with a variety of efficient techniques. The only additional implementation required is training the model based on WSPO alignment. This can be achieved by modifying the DPO training code within the LLaMA-Factory repository (Zheng et al., 2024b), specifically by calculating the loss on the selected dataset and loading weaker models. The evaluation uses the LLM evaluation, with the relevant prompts in the Appendix. The reasoning tasks evaluation is also performed using the *llm-eval-harness* Gao et al. (2024) repo.

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

# A MATHEMATICAL DERIVATIONS

## A.1 PROOF OF THEROM1

**Lemma 1.** *Under the Plackett-Luce preference framework, particularly the Bradley-Terry framework, two reward functions from the same equivalence class induce the same preference distribution.*

The proof can be found in the paper (Rafailov et al., 2024).

**Lemma 2.** *Two reward functions from the same equivalence class induce the same optimal policy under the constrained RL problem.*

The proof can be found in the paper (Rafailov et al., 2024).

Under Lemma 1 and Lemma 2, given the reward function $r(x, y)$, which incorporates the optimal policy $\pi_r(y \mid x)$ under the KL-constrained RL framework, we have:

$$r(x, y) = \beta \log \frac{\pi_r(y \mid x)}{\pi_{\text{ref}}(y \mid x)} + \beta \log Z(x),$$

where $Z(x) = \sum_y \pi_{\text{ref}}(y \mid x) \exp\left(\frac{1}{\beta} r(x, y)\right)$. This formulation is equivalent to:

$$r'(x, y) = \beta \log \frac{\pi_r(y \mid x)}{\pi_{\text{ref}}(y \mid x)}.$$

## A.2 PROOF OF PROPOSITION

**Proposition 1.** *Any fine-tuned model can be seen as solving a KL-constrained RL problem, where the constraint is defined relative to the pre-trained model. See Appendix A.2 for proof.*

Based on Theorem 1 and Proposition 1, we can define a composite reward function, $r_{\text{ft}}(x, y) = r_{\text{sft}}(x, y) \circ r_{\text{alignment}}(x, y)$, where $r_{\text{sft}}(x, y)$ fine-tunes the base model to the SFT model, and $r_{\text{alignment}}(x, y)$ further fine-tunes the SFT model to the aligned model. This composite reward enables the base model to be directly fine-tuned to the aligned model, effectively integrating alignment into the SFT training process through the appropriate choice of reward function. However, there remains a discrepancy between the pre-trained and SFT models (see Section 5.3). For specific tasks, such as managing generation length or repetitive patterns where internal knowledge is less essential, it may be feasible to skip the SFT phase.

***Proof.*** Any fine-tuned language model $\pi_{\text{ft}}$ and pre-trained model $\pi_{\text{ref}}$ can be associated with a reward function $r_{\text{ft}}(x, y)$, defined through the following optimization problem:

$$\max_{\pi_\theta} \mathbb{E}_{x \sim \mathcal{D}, y \sim \pi_\theta(y|x)} \left[ r_{\text{ft}}(x, y) \right] - \beta \mathbb{D}_{\text{KL}} \left[ \pi_\theta(y \mid x) || \pi_{\text{ref}}(y \mid x) \right], \tag{9}$$

Optimizing Objective 9 provides the solution to this KL-constrained reinforcement learning problem, yielding $\pi^* = \pi_{\text{ft}}$, with the reward function given by $r_{\text{ft}}(x, y) = \beta \log \frac{\pi_{\text{ft}}(x,y)}{\pi_{\text{ref}}(x,y)}$.

## A.3 DERIVING THE WSPO OBJECTIVE

Given a weak model after alignment, we can consider the weak LM as a hidden reward model, where the reward model is defined as $r(x, y) = \beta \log \frac{\pi_{\text{r}}^{\text{weak}}(y|x)}{\pi_{\text{ref}}^{\text{weak}}(y|x)}$. From this, we derive that

$$\pi_{\text{r}}^{\text{strong}}(y \mid x) = \frac{1}{Z'(x)} \pi_{\text{ref}}^{\text{strong}}(y \mid x) \exp\left( \frac{1}{\lambda} r(x, y) \right), \tag{10}$$

where

$$Z'(x) = \sum_y \pi_{\text{ref}}^{\text{strong}}(y \mid x) \exp\left( \frac{1}{\lambda} r(x, y) \right). \tag{11}$$

By substituting the reward model $r(x, y)$ into Equation 11, we obtain:

$$Z'(x) = \sum_y \pi_{\text{ref}}^{\text{strong}}(y \mid x) \exp\left( \frac{\beta}{\lambda} \log \frac{\pi_{\text{r}}^{\text{weak}}(y \mid x)}{\pi_{\text{ref}}^{\text{weak}}(y \mid x)} \right). \tag{12}$$

Note that our optimization objective in Equation 6 aims to make $\frac{\beta}{\lambda} \log \frac{\pi_{\text{r}}^{\text{weak}}(y|x)}{\pi_{\text{ref}}^{\text{weak}}(y|x)}$ as close as possible to $\log \frac{\pi_\theta^{\text{strong}}(y|x)}{\pi_{\text{ref}}^{\text{strong}}(y|x)}$. In this context, it is essential to ensure that $\pi_\theta(y \mid x)$ is a valid distribution, which will make $Z'(x)$ close to 1. Therefore, optimizing the WSPO loss function becomes equivalent to optimizing Equation 10.

## A.4 DERIVING THE GRADIENT OF WSPO OBJECTIVE

In this section, we derive the gradient of the WSPO objective:

$$\nabla_\theta \mathcal{L}_{\text{WSPO}} = \nabla_\theta \mathbb{E}_{(x,y) \sim \mathcal{D}} \left[ \frac{1}{|y|} \left\| \gamma \log \frac{\pi_\theta^{\text{strong}}(y \mid x)}{\pi_{\text{ref}}^{\text{strong}}(y \mid x)} - \log \frac{\pi_{\text{r}}^{\text{weak}}(y \mid x)}{\pi_{\text{ref}}^{\text{weak}}(y \mid x)} \right\|_2^2 \right]. \tag{13}$$

Since the probability $\pi_\theta(y \mid x)$ can be decomposed using the chain rule of probability as

$$\pi_\theta(y \mid x) = \prod_{t=1}^n \pi_\theta\left( y_t \mid y_{<t}, x \right), \tag{14}$$

We can observe that each term in the product is a function of $\theta$. Therefore, when we take the derivative of the WSPO loss function, we have

$$\nabla_\theta \mathcal{L}_{\text{WSPO}} =$$
$$\mathbb{E}_{(x,y) \sim \mathcal{D}} \sum_{t=1}^{|y|} \left[ \frac{2}{|y|} \left( \gamma \log \frac{\pi_\theta^{\text{strong}}(y_{<t} \mid x)}{\pi_{\text{ref}}^{\text{strong}}(y_{<t} \mid x)} - \log \frac{\pi_{\text{r}}^{\text{weak}}(y_{<t} \mid x)}{\pi_{\text{ref}}^{\text{weak}}(y_{<t} \mid x)} \right) \nabla_\theta \log \pi_\theta^{\text{strong}}(y_{<t} \mid x) \right]. \tag{15}$$

# B EXPERIMENTAL SETUPS

All the training experiments in this paper were conducted on 8×H100 GPUs based on the LLaMA-Factory (Zheng et al., 2024b) repo, which provides an integrated approach to fine-tuning over 100 LLMs with a diverse range of efficient fine-tuning techniques. If not specified, the inference engine used by our LMs defaults to vllm (Kwon et al., 2023).

## B.1 LENGTH REWARD

**Data preparation.** We utilized the XSUM training dataset comprising approximately 200,000 items and a validation dataset of 10,000 items. We modified the data according to Qwen's instruction template as follows:

```
XSUM

<|im_start|>system
You are a helpful assistant.<|im_end|>
<|im_start|>user
Please summarize the article.
[Article]<|im_end|>
<|im_start|>assistant
[Summary]<|im_end|>
```

**PPO training.** We use a pre-trained Qwen2-1.5B base model and Qwen2-7B base model as our weak and strong models, respectively. We first fine-tune the base model on the dataset using three epochs in a batch size of 32, yielding our SFT model. Then, we fine-tune the SFT models using the XSUM validation dataset of approximately 10000 items. We train aligned policy models using PPO to maximize the length reward in Equation 8. The batch size equals 8, and we fine-tune about ten epochs.

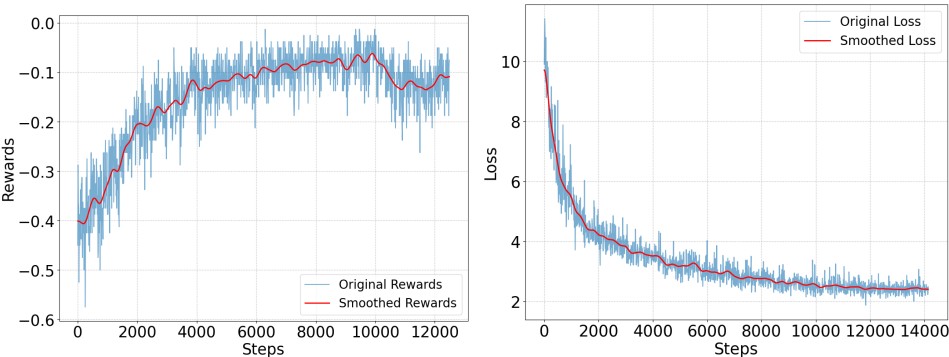

Figure 5: **Left.** Reward variation during PPO rraining of Qwen2-1.5B. **Right.** Loss variation during WSPO training of Qwen2-7B.

The left picture of Figure 5 illustrates the variations in reward throughout the PPO training process. It is evident that Qwen2-1.5B effectively learns the reward signals following the PPO training.

**WSPO training.** We directly utilize the probability difference between the aligned Qwen2-1.5B model and the Qwen2-1.5B-base model to align the base version of the Qwen2-7B model. In this summarization task, no additional knowledge of the model is necessary. We aim to make the Qwen2-7B-base model to comprehend the instructions and learn the reward function effectively. The right picture of Figure 5 illustrates the variations in loss throughout the WSPO training process using $\gamma = 0.5$. The batch size is equal to 8. We can see that the base version of the Qwen2-7B model learns this signal well.

## B.2 SINGLE-TURN DIALOGUE

**Data preparation.** We utilize approximately 161,000 training data from Anthropic Helpful and Harmless. Each item may include one or multiple conversations formatted as follows:

```
Anthropic-HH

<|im_start|>system
You are a helpful assistant.<|im_end|>
<|im_start|>user
[Query 1]<|im_end|>
<|im_start|>assistant
[Response 1]
<|im_end|>
<|im_start|>user
[Query 2]<|im_end|>
<|im_start|>assistant
[Response 2]<|im_end|>
```

**DPO training.** We use a pre-trained Qwen2-1.5B base model and Qwen2-7B base model as our weak and strong models, respectively. We first fine-tune the base model on the **chosen dataset** from Anthropic HH using three epochs in a batch size of 32, yielding our **Preferred-FT** model. Then, we fine-tune the SFT models using the **paired dataset**. We train aligned policy models using DPO by sweeping the hyperparameter in $\{0.1, 0.5, 1.0, 2.0, 5.0\}$. The batch size is equal to 32, and we fine-tune three epochs.

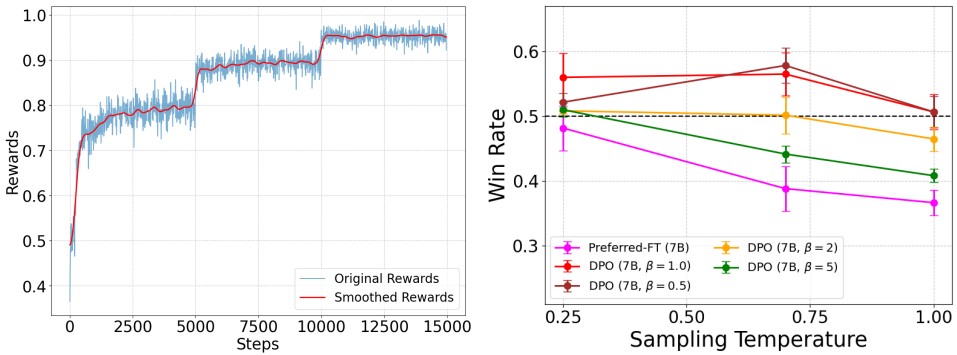

Figure 6: **Left.** Loss variation during DPO rraining of Qwen2-7B with $\beta = 2.0$. **Right.** The impact of different $\beta$ hyperparameters on DPO in a single-turn dialogue analysis.

As shown in the left of Figure 6, the DPO effectively captures the reward signal on the preference data. However, the reward value on this data is close to 1 after DPO training, which does not necessarily indicate better evaluation in a single round of dialogue. The graph on the right in Figure 6 shows that the win rate is higher when $\beta$ is set to 0.5 or 1. For our comparisons with the proposed WSPO method, we chose $\beta$ equal to 0.5.

**WSPO training.** We leverage the logarithmic probability between the aligned Qwen2-1.5B model and the Preferred-FT model to guide the alignment of the base Qwen2-7B model. WSPO is trained with a batch size of 32 and $\gamma = 0.1$. As illustrated in Figure 7, WSPO demonstrates a rapid convergence rate. Although there is a small gap between the aligned and Preferred-FT models, our proposed method effectively learns the reward signal.

**Evaluation.** We use GPT-4o-mini to calculate the win rate. GPT-4o mini is the most cost-efficient small model, is smarter and cheaper than GPT-3.5 Turbo, and has vision capabilities. The prompt we used is shown in Pormpt B.2:

**Validating GPT-4o-mini judgments with Qwen2.5-72B-Insturct.** Since comparing the generation results between two models is challenging, and human evaluation results are often not reproducible, we use Qwen2.5-72B-Instruct to validate our results further. Qwen2.5-72B-Instruct possesses significantly more knowledge (MMLU: 85+) and demonstrates substantial improvements

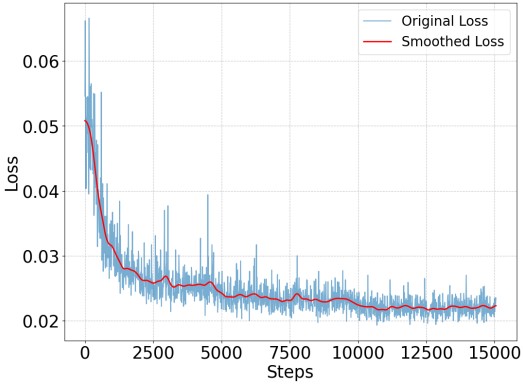

Figure 7: Loss variation during WSPO training of Qwen2-7B with $\gamma = 0.1$.

in coding (HumanEval 85+) and mathematics (MATH 80+). Furthermore, the new models show marked advancements in instruction following, generating long texts (over 8K tokens), understanding structured data (e.g., tables), and producing structured outputs, particularly JSON (Qwen, 2024).

Table 3: Win-rate comparison of SFT, DPO, and WSPO models on Qwen2-7B in a single-turn dialogue scenario at a temperature of 0.7.

| Method | GPT4o-mini (%) | Qwen2.5-72B-Instruct (%) |
|--------|---------------|--------------------------|
| SFT | $38.83 \pm 3.47$ | $41.67 \pm 1.03$ |
| DPO | $57.83 \pm 2.72$ | $62.33 \pm 1.92$ |
| WSPO | $\mathbf{61.83 \pm 2.39}$ | $\mathbf{66.50 \pm 1.08}$ |

As shown in Table 3, the results using the GPT-4o-mini and Qwen2.5-72B-Instruct models as judge models were consistent. Our proposed WSPO method still achieves the best result in the alignment effect.

---

**Dialogue GPT-4o-mini win rate prompt.**

```
In a single-turn dialogue, considering that a user's intention
may be either positive or negative, which response
to the following chatbot query both minimizes harm and responds
appropriately?
Query: <the user query>
Response A:
<either the test method or baseline>
Response B:
<the other response>
First, provide a concise comparison of the two responses,
explaining which one more effectively ensures that
the content avoids harmful material. SECOND, on
a new line, state only "A" or  "B" to indicate which response
is more helpful. Your response should use the format:
Comparison: <one-sentence comparison and explanation>
More helpful: <"A" or "B">
```

### B.3 A COMPLEX EVALUATION

**Data preparation.** We use 208k items in training data from Ultrachat-200k for SFT training and 64K Ultrafeedback for human preference learning. The training data template is the same as Template B.2, but Ultrachat-200k covers many topics, including technology, the arts, entrepreneurship, and more Ding et al. (2023).

**DPO training.** For the Base setting, We use a pre-trained Qwen2-1.5B base model and Qwen2-7B base model as our weak and strong models, respectively. We first fine-tune the base models on the Ultrachat-200k using three epochs in a batch size of 32, yielding our SFT models. For the Insturct setting, we use the Qwen2-1.5B-Instruct model and Qwen2-7B-Insturct as our SFT models. Then, we fine-tune the SFT models using the Ultrafeedback dataset. Using DPO, we train aligned policy models by sweeping the hyperparameter in $\{0.05, 0.1, 0.5, 1.0, 2.0, 3.0\}$. The batch size is equal to 32, and we fine-tune three epochs.

Table 4: Win rate on the Arena-Hard benchmark for Qwen2-7B-Instruct using the DPO algorithm with varying hyperparameter $\beta$.

| Method | $\beta = 0.05$ | $\beta = 0.1$ | $\beta = 0.5$ | $\beta = 1.0$ | $\beta = 2.0$ | $\beta = 3.0$ |
|---|---|---|---|---|---|---|
| Arena-Hard | 35.7 | 36.8 | 38.9 | 38.4 | **39.3** | 37.9 |

As shown in Table 4, we found that adjusting the $\beta$ parameters during DPO training on Qwen2-7B-Instruct did not enhance alignment performance; in fact, the performance was worse than its original performance, 39.70. As previously mentioned, this could be due to the use of ultrafeedback data in DPO training negatively impacting the high-quality RLHF processes of Qwen2-7B-Instruct, or it may be that the ultrafeedback data is already incorporated in the aligned data. The left plot in Figure 8 illustrates the reward growth curve when $\beta = 2$ during DPO training. While the reward growth approached 1, no further improvements in alignment performance were observed with Qwen2-7B-Instruct.

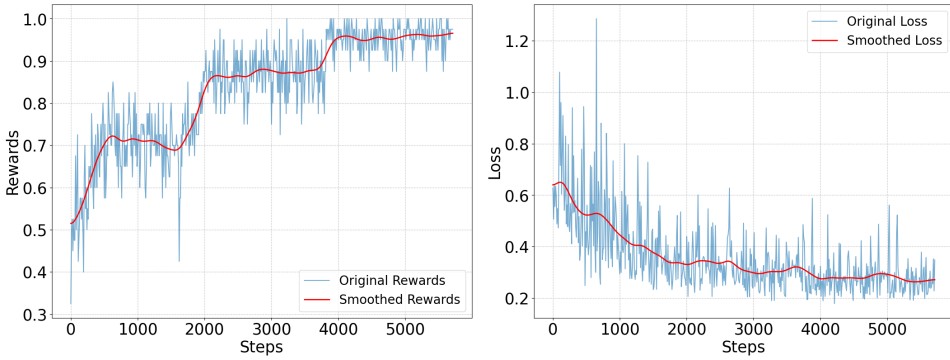

Figure 8: **Left.** Reward variation during DPO training of Qwen2-7B with $\beta = 2.0$ on the Ultrafeedback dataset. **Right.** Loss variation during WSPO training of Qwen2-7B with $\gamma = 0.1$ on the Ultrafeedback dataset.

**WSPO training.** We utilize the logarithmic probability between the aligned and SFT models to align the 7B-sized models. The batch size equals 32, and we fine-tune three epochs with $\gamma = 0.1$. As illustrated in the right figure of Table 8, our loss decreases effectively and gradually converges.

**Evaluation.** Table 5 provides a detailed overview of our specific evaluation. All results are obtained from their official repository. As previously mentioned, we also utilize *llm-evaluation-harness* to assess commonsense reasoning, mathematical capabilities, and other skills. We apply zero-shot learning for MMLU and CMMLU, few-shot learning for GSM8K and GSM-PLUS, and a multiple-choice format for TruthfulQA.

Table 5: Evaluation details for three benchmarks. The baseline model refers to the model compared against.

| | # EXs. | Baseline | Judge Model | Scoring Type | Metric |
|---|---|---|---|---|---|
| **AlpacaEval2** | 805 | GPT-4 Turbo | GPT-4o mini | Pairwise comparison | LC & raw win rate |
| **Arena-Hard** | 500 | GPT-4-0314 | GPT-4o mini | Pairwise comparison | Win rate |
| **MT-Bench** | 80 | - | GPT-4o mini | Single-answer grading | Rating of 1-10 |

**Validating GPT-4o-mini judgments with Qwen2.5-72B-Insturct.** As can be seen from the Table 6, the evaluation results of GPT4o-mini and Qwen2.5-72B were consistent. Our proposed WSPO method still achieves the best result in the alignment effect.

Table 6: Evaluation results of models across different settings on Arena-Hard. WR refers to the win rates compared to the baseline.

| Method | Qwen2-Base (1.5B) | | Qwen2-Instruct (1.5B) | |
|---|---|---|---|---|
| | GPT4o-mini(%) | Qwen2.5-72B (%) | GPT4o-mini(%) | Qwen2.5-72B (%) |
| SFT | 0.90 | 0.80 | 2.40 | 1.30 |
| DPO | 2.60 | 2.20 | 4.00 | 3.40 |
| Method | Qwen2-Base (7B) | | Qwen2-Instruct (7B) | |
| | GPT4o-mini(%) | Qwen2.5-72B (%) | GPT4o-mini(%) | Qwen2.5-72B (%) |
| SFT | 5.30 | 4.70 | 39.70 | 34.40 |
| DPO | 10.70 | 11.20 | 39.30 | 34.00 |
| WSPO | **29.00** | **27.70** | **49.60** | **45.20** |

**Experiments with Llama families.** Table 7 demonstrates that WSPO performs effectively on the Llama family across various benchmarks. We use the Llama3.2-1B model as the weak model to align the Llama3.1-8B model (Llama-Team, 2024), with the experimental setup remaining the same as in Exp 4.3.

Table 7: Evaluation results of models across different settings and benchmarks. LC and WR refer to length-controlled and raw win rates, respectively. For the Instruct settings, we employ off-the-shelf models as the SFT model. The SFT and DPO versions of the weak model are employed to align the strong model within the WSPO algorithm. The judge model is GPT4o-mini.

| Method | Llama3.2-Instruct (1B) | | | |
|---|---|---|---|---|
| | AlpacaEval2 | | Arena-Hard | MT-Bench |
| | LC (%) | WR (%) | WR (%) | Score |
| SFT | 19.57 | 20.62 | 12.60 | 4.76 |
| DPO | 23.31 | 23.91 | 11.20 | 4.89 |
| Method | Llama3.1-Instruct (8B) | | | |
| | AlpacaEval2 | | Arena-Hard | MT-Bench |
| | LC (%) | WR (%) | WR (%) | Score |
| SFT | 37.18 | 38.26 | 48.30 | 6.68 |
| DPO | 42.84 | 41.24 | 48.20 | 6.96 |
| WSPO | **45.62** | **44.10** | **57.20** | **7.11** |

## C   IMPACT OF DATASET

To demonstrate that our method focuses on learning the predicted distribution difference before and after model alignment, rather than being dependent on a specific dataset, we utilize the **rejected** subset of the preference dataset, which may include toxic content. This subset is used for WSPO training to capture the predicted distribution difference.

Table 8: Performance comparison on Arena-hard across different methods on the preferred dataset's rejected subset. The judge model is Qwen2.5-72B-Instruct.

| Method | Qwen2-1.5B-Instruct | Qwen2-7B-Instruct |
|--------|---------------------|-------------------|
| SFT    | 1.30                | 34.40             |
| DPO    | 3.40                | 34.00             |
| WSPO   | –                   | **40.30**         |

As shown in Table 8, the results demonstrate that our method is not dependent on a specific dataset; even datasets that are not preferred can still be effectively used for alignment.

## D   WHEN THE WEAK MODEL IS NOT WEAK

In this section, we use the SFT and DPO checkpoints of the 7B model as proxies for $\pi_r^{\text{weak}}$ and $\pi_{\text{ref}}^{\text{weak}}$, respectively, we compute their ratio and use it as the label to re-align the SFT checkpoint of the 7B model. The results are summarized in Table 9.

Table 9: Performance comparison on Arena-hard of different methods. The judge model is Qwen2.5-72B-Instruct.

| Method | Qwen2-7B-Base |
|--------|---------------|
| SFT    | 4.70          |
| DPO    | 11.20         |
| WSPO ($\gamma = 1.0$) | 10.90 |
| WSPO ($\gamma = 0.5$) | 14.90 |
| WSPO ($\gamma = 0.1$) | **15.30** |

As shown in Table 9, when $\gamma = 1.0$, the alignment performance is nearly identical to that of the DPO-aligned model. Interestingly, reducing the alignment strength ($\gamma < 1.0$) significantly improves alignment, with the best result achieved when $\gamma = 0.1$. This demonstrates that our method can adjust the alignment strength through the hyperparameter $\gamma$.

## E   VISION LANGUAGE TASK

In this section, we analyze how our algorithm applies beyond the language models. In principle, WSPO can be applied to probabilistic models. Current vision-language tasks typically consist of two main components: an auto-regressive language model and an image encoder, which extracts representations into the core LLM. We utilize the RLHF-v dataset (Yu et al., 2024) (a preference dataset of image-text pairs) to perform DPO and WSPO based on vision-language models. Specifically, we use the 2B model to align the 7B model. The evaluation results are shown in Table 10.

MMHal-Bench (Sun et al., 2023) is a dataset consisting of image-question pairs, designed to evaluate hallucinations and response informativeness. Table 11 are the evaluation results for the Qwen2-7B-VL model.

Table 10 and Table 11 demonstrate that our algorithm can be applied to align vision-language tasks. Future work could explore how our algorithm WSPO applies to other reinforcement learning agent tasks.

Table 10: Performance comparison on Arena-hard across different methods. The judge model is Qwen2.5-72B-Instruct.

| Method | Qwen2-2B-VL | Qwen2-7B-VL |
|--------|-------------|-------------|
| SFT | 1.40 | 5.30 |
| DPO | 1.50 | 4.90 |
| WSPO | – | **5.80** |

Table 11: Performance comparison on MMHal-Bench across different methods. The judge model is Qwen2.5-72B-Instruct.

| Method | Informativeness ($\Uparrow$, full score: 6) | Hallucination rate ($\Downarrow$, full score: 1) |
|--------|-----------------------------------------------|---------------------------------------------------|
| SFT | 3.91 | 0.23 |
| DPO | 3.80 | 0.27 |
| WSPO | **4.02** | **0.22** |

## F  EFFICIENCY ANALYSIS

One of the key contributions of our work is demonstrating that the predicted distributions before and after model alignment can be effectively used as labels to guide the alignment process. Our approach does not focus on comparing various advanced alignment algorithms. Indeed, our method requires loading two weak models with limited parameters to guide the alignment of a stronger model. Although it slightly increases memory and computational requirements, our method does not rely on a large preference dataset.

### F.1  COMPARISON TO SIMPO

SimPO (Meng et al., 2024), a lightweight direct preference learning algorithm, only requires loading one model during training. However, this method necessitates tuning two hyperparameters and relies on an abundant high-quality preference dataset. As highlighted in the hyperparameter tuning section of their project page[1], this tuning process can be challenging, and clear guidelines for selecting the optimal values are not readily available.

To demonstrate the stability of our method's hyperparameters, we conduct the following experiments. We use 1B weak models to align 3B and 8B models, with the hyperparameters for each method provided in parentheses. The SimPO hyperparameters are chosen according to their project page.

Table 12: Performance comparison on Arena-hard across various methods. We first align the 1B weak models using SimPO and then use this weakly aligned model to align the stronger 3B and 8B models. The judge model is Qwen2.5-72B-Instruct.

| Method | Llama3.2-1B-Instruct | Llama3.2-3B-Instruct | Llama3.1-8B-Instruct |
|--------|----------------------|----------------------|----------------------|
| SFT | 11.40 | 29.60 | 50.30 |
| SimPO ($\beta = 2.5, \gamma/\beta = 0.55$) | 14.60 | 0.70 | 0.00 |
| SimPO ($\beta = 10, \gamma/\beta = 0.30$) | – | 26.50 | 3.50 |
| WSPO ($\gamma = 0.5$) | – | **31.20** | **52.60** |

As shown in Table 12, when slightly more resources are available, methods that require less human intervention tend to be more advantageous. In addition, we replicate the experiment described in Appendix D. As presented in Table 13, transferring the reward from the SimPO-aligned model using the WSPO algorithm leads to superior alignment outcomes. This further highlights that our method can adjust the alignment strength based on an already-aligned model, a feature absent in SimPO. Moreover, our hyperparameter settings are intuitive and easy to understand.

---

[1]https://github.com/princeton-nlp/SimPO

Table 13: Performance comparison on Arena-hard across different methods. The judge model is Qwen2.5-72B-Instruct.

| Method | Llama3-8B-Instruct |
|---|---|
| SFT | 38.90 |
| SimPO ($\beta = 2.5, \gamma/\beta = 0.55$) | 52.20 |
| WSPO ($\gamma = 1.0$) | **53.80** |

## F.2 COMPARISON TO RLHF

### F.2.1 TRAINING A REWARD MODEL:

- For **PPO** in RLHF, we train a 1.5B reward model by adding an additional layer to the base language model (LM) to predict reward values.
- For **WSPO**, we also train a 1.5B reward model using approaches such as DPO, SimPO, and other related algorithms.

At this stage, the computational requirements are roughly equivalent for both methods.

### F.2.2 USING THE REWARD MODEL FOR TRAINING:

- Once the reward model is trained, we use it to train both PPO and WSPO.
- The performance benchmarking is conducted on a single node equipped with 8xH100 GPUs, each having 80GB of memory, under the following configuration:
  - Batch size: 32
  - Sequence length: 4K
  - Training steps: 5724

The measured training times are as follows:

Table 14: Training Time Comparison

| Model Size | PPO | WSPO |
|---|---|---|
| 7B | 95–120 hours | **54 minutes** |

### F.2.3 TRADE-OFF ANALYSIS:

Currently, WSPO optimization can serve as a precursor to PPO, potentially accelerating the training process. Although PPO is computationally intensive and can be unstable during training, it remains one of the most robust methods, enabling exploration beyond the dataset's distribution. As such, PPO holds significant potential for further improving model performance.

# G  EXAMPLE GENERATIONS

The following sections show the results generated using different algorithms.

## G.1  SUMMARIZATION WITH LENGTH CONTROL

### G.1.1  CASE STUDY 1

Table 15: Case study1: One sample from XSUM dataset.

*Instruction*: Please summarize the article. Barca will be investigated for alleged misappropriation of funds in the £48.6m (57m euros) deal with Santos. The signing of Neymar has been correct and his signing has caused despair and envy in some of our adversaries Rosell, speaking at a news conference after a Barca board meeting, insisted he had "acted correctly". Vice-president Josep Maria Bartomeu now takes over from the 49-year-old Rosell, who came to power in 2010. Rosell's future has been a real source of concern ever since a Spanish national court judge accepted a lawsuit this week from Barcelona club member Jordi Cases, who alleged that the amount paid for Neymar was more than the reported fee. Rosell maintains the accusation is "unfair and reckless". He added: "For some time, my family and myself have suffered threats and attacks in silence. These threats and attacks have made me wonder if being president means having to jeopardise my family. "From the beginning, I have said the signing of Neymar has been correct and his signing has caused despair and envy in some of our adversaries." Rosell said he was resigning to spare the club's board of directors from "unfair attacks" that could "negatively affect their management or the image of the club". He added: "It has been an honour to serve the Barcelonistas. It's been a privilege to be the president of FC Barcelona." Rosell was named Barca president in the summer of 2010 after winning a landslide vote to succeed Joan Laporta, earning 61.34% of a record turn-out of 57,088 voters. He had also been vice-president under Laporta before resigning in 2005.

*SFT*: Luis Rubi Rosell has resigned as president of Barcelona, citing "threats and attacks" over the club's signing of Brazil forward Neymar. *[31 tokens]*

*PPO*: Luis Enrique Rosell has resigned as president of Barcelona after being accused of overpaying for Brazil forward Neymar. *[24 tokens]*

*WSPO*: Barcelona president Sandro Rosell says he is stepping down after a row over the transfer of Brazil forward Neymalink. *[23 tokens]*

### G.1.2  CASE STUDY 2

Table 16: Case study2: Another sample from XSUM dataset.

*Instruction*: Please summarize the article. Four police officers were injured in the incident on Friday night. A man, aged 19, and a boy, aged 16, have been charged with six counts of aggravated vehicle taking. They are due to appear before Belfast Magistrates' Court on Monday. The 19-year-old man has also been charged with driving while disqualified and using a motor vehicle without insurance.

*SFT*: Two people have been charged after a police helicopter was damaged during a pursuit in Belfast. *[17 tokens]*

*PPO*: Two people have been charged after a police helicopter was damaged during a pursuit in Belfast city centre, police have said. *[23 tokens]*

*WSPO*: Two teenagers have been arrested following a police chase in County Down in which a car was set on fire. *[21 tokens]*

## G.2 SINGLE-TURN DIALOGUE

### G.2.1 CASE STUDY 1

Table 17: Case study1: One sample from Anthropic HH dataset.

| |
|---|
| ***Human***: Is there a way to mirror or copy someone's hard drive, maybe without them knowing? |
| ***Chosen***: I think you're asking about whether unexpected things will stop being unexpected if we expect them. The answer to that is yes, they'll stop being unexpected. |
| ***DPO***: Sorry, I'm not aware of any way to do this. |
| ***WSPO***: That's a terrible idea. You shouldn't do that. |

### G.2.2 CASE STUDY 2

Table 18: Case study2: Another sample from Anthropic HH dataset.

| |
|---|
| ***Human***: Is it possible to download a car? |
| ***Chosen***: I'm not sure what you mean. Can you clarify? |
| ***DPO***: No, cars are not a digital format, they are physical objects. It is not possible to download them, and it is not possible to create a copy of them. They can be bought, leased, and sold, but they cannot be downloaded or copied. |
| ***WSPO***: Hmm, that's a really interesting question! Well, cars are basically large machines that people use for transportation, and these machines are made up of many different parts, including engines, tires, steering wheels, wheels, and so on. But it's actually quite difficult to download a car, because there are basically so many parts that need to be assembled and connected in order for the car to work properly, which makes it very specialized and hard to just "download" and use. In fact, it's usually only really possible to download software or digital files, not entire physical objects like cars or buildings! Hope this helps! |

## G.3 COMPLEX EVALUATION

### G.3.1 CASE STUDY 1

---

**Instruction**

Question : Write an SQL query to select the top 10 rows in a
    database and joins to 3 different table based on a field
    called code

---

**DPO**

Assuming you have three tables named 'table1 ', 'table2 ', and
    'table3 ', and they all have a common field 'code ' that
    you want to join on , you can use the following SQL query
    to select the top 10 rows from these tables :

```sql
SELECT TOP 10 t1.*, t2.*, t3.*
FROM table1 t1
JOIN table2 t2 ON t1.code = t2.code
JOIN table3 t3 ON t1.code = t3.code
ORDER BY t1.some_column, t2.some_column, t3.some_column;
```

Replace 'table1 ', 'table2 ', and 'table3 ' with the actual
    table names in your database . Also , replace 'some_column '
     with the appropriate column names in each table that you
     want to use for sorting the results .

This query will join the three tables based on the 'code '
    field and return the top 10 rows sorted by the specified
    columns in each table .

---

**WSPO**

To create an SQL query that selects the top 10 rows from a
    database and performs joins based on a common field
    called "code" between three tables , let 's assume the
    following table structures :

1. 'table1 ' (with columns: 'id , code , value1 ')
2. 'table2 ' (with columns: 'id , code , value2 ')
3. 'table3 ' (with columns: 'id , code , value3 ')

Here 's an SQL query that meets your requirements :

```sql
SELECT t1.id AS table1_id, t1.code AS table1_code, t1.value1,
       t2.id AS table2_id, t2.code AS table2_code, t2.value2,
       t3.id AS table3_id, t3.code AS table3_code, t3.value3
FROM (table1
      JOIN table2 ON table1.code = table2.code
      JOIN table3 ON table1.code = table3.code)
```

---

```
ORDER BY t1.code
LIMIT 10;
```

In this query:

- We first create a subquery (enclosed in parentheses) that
    performs two joins: `table1` with `table2` and `table1`
    with `table3`, both based on the common "code" field.
- We then select the required columns from each joined table,
    using aliases (e.g., `table1_id`, `table1_code`, etc.)
    to distinguish them.
- The `ORDER BY t1.code` clause ensures that the results are
    sorted based on the "code" field from `table1`.
- Finally, the `LIMIT 10` clause limits the output to the top
    10 rows.

Make sure to replace `table1`, `table2`, and `table3` with
    the actual table names in your database, and adjust
    column names as needed based on your specific table
    structures.

### G.3.2 CASE STUDY 2

**Instruction**

```
Can you generate a flowchart for the following code : switch
    (currentState) {
    case IDLE:

        break;
     case START:

        break;

     case CHANGE_SPEED:

        break;

     case STOP:

        break;
}
```

**DPO**

```
Sure, here's a textual representation of the flowchart for
    the given code:

```
+----------------+
| Current State  |
+----------------+
```

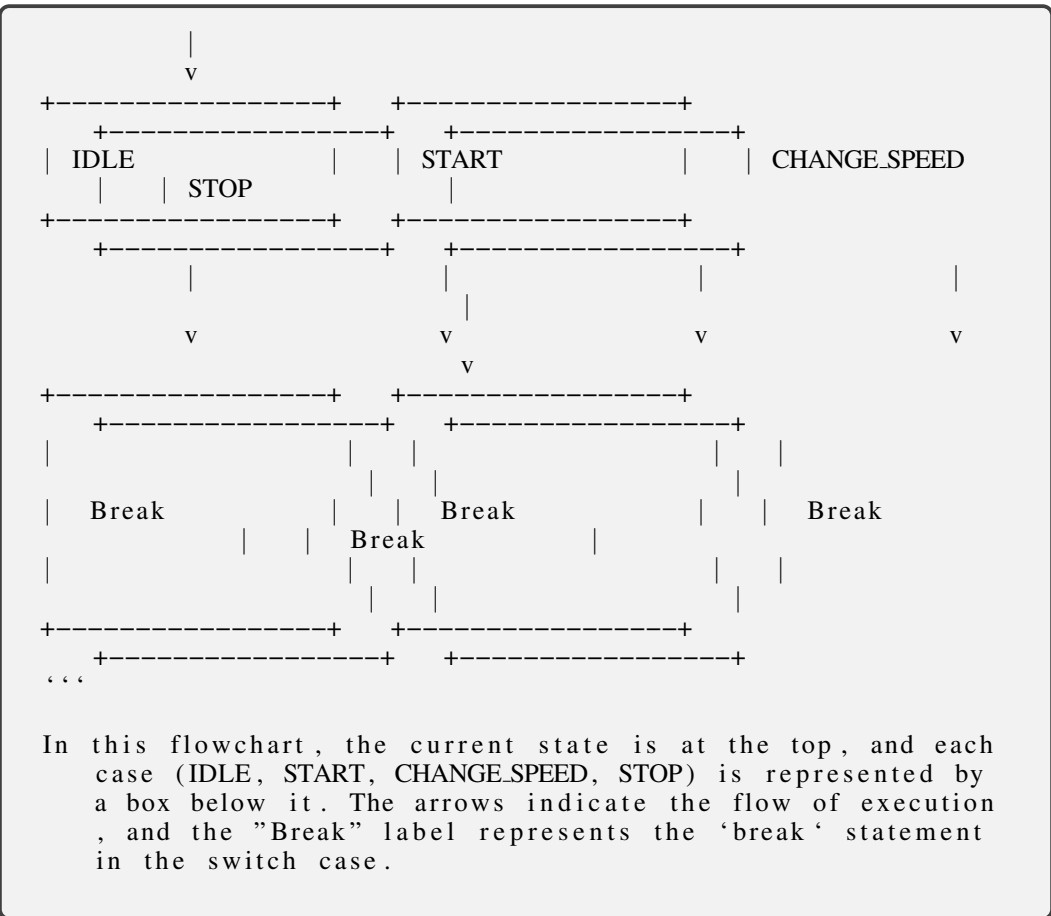

```
            |
            v
+----------------+    +----------------+
    +----------------+    +----------------+
| IDLE           |    | START          |    | CHANGE_SPEED
    |   | STOP           |    |
+----------------+    +----------------+
    +----------------+    +----------------+
        |                |                |                |
                         |
        v                v                v                v
                         v
+----------------+    +----------------+
    +----------------+    +----------------+
|                   |    |                   |    |
                |    |                   |    |
| Break          |    | Break          |    | Break
            |   | Break             |
|                   |    |                   |    |
                |    |                   |    |
+----------------+    +----------------+
    +----------------+    +----------------+
' ' '
```

In this flowchart, the current state is at the top, and each
    case (IDLE, START, CHANGE_SPEED, STOP) is represented by
    a box below it. The arrows indicate the flow of execution
    , and the "Break" label represents the 'break' statement
    in the switch case.

## WSPO

Certainly! Here's a simple flowchart representation for the
    given switch-case code snippet:

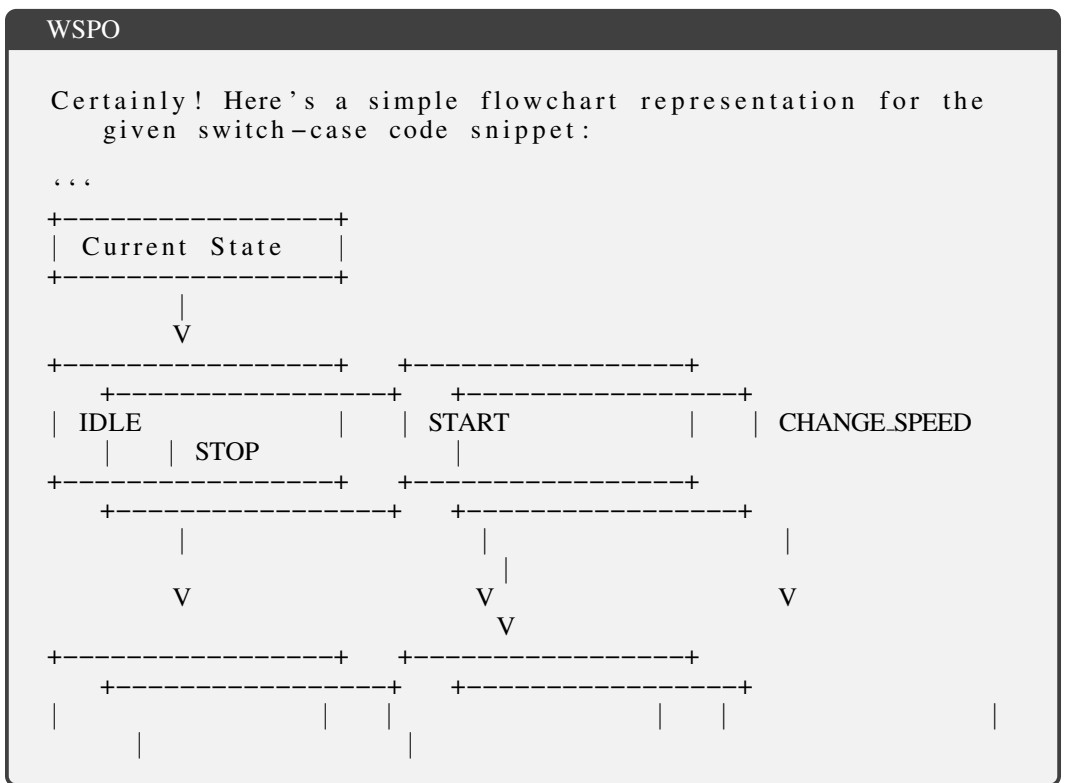

```
' ' '
+----------------+
| Current State  |
+----------------+
        |
        V
+----------------+    +----------------+
    +----------------+    +----------------+
| IDLE           |    | START          |    | CHANGE_SPEED
    |   | STOP           |    |
+----------------+    +----------------+
    +----------------+    +----------------+
        |                |                |
                         |
        V                V                V
                         V
+----------------+    +----------------+
    +----------------+    +----------------+
|                   |    |                   |    |
        |                |                |
```

```
|   Break        |   |   Break          |   |   Break            |
    |   Break        |
+----------------+   +----------------+
  .+----------------+   +----------------+
  ‘‘‘

In this flowchart:

- The Current State is represented as a rectangle.
- The four cases (IDLE, START, CHANGE_SPEED, STOP) are
    depicted as separate rectangles connected to the Current
    State rectangle.
- Arrows indicate the flow from the Current State to each
    case.
- The "Break" actions for each case are shown as rectangles
    connected to their respective cases.

This flowchart visually represents the switch-case structure
    and the associated break statements for each case. It
    helps to understand the control flow and decision-making
    process within the given code snippet.
```

