# OpenReview forum: "Weak-to-Strong Preference Optimization: Stealing Reward from Weak Aligned Model"
_ICLR.cc/2025/Conference — ICLR 2025 Spotlight_

### Official Review · Reviewer_ssa6 · 2024-10-27

**Soundness:** 3
**Presentation:** 2
**Contribution:** 3
**Rating:** 8
**Confidence:** 3

**Summary:**

This paper introduces Weak-to-Strong Preference Optimization (WSPO), a novel method for aligning large language models (LLMs) with human preferences. Inspired by the concept of weak-to-strong generalization, WSPO leverages distribution shifts observed during the alignment of weaker models to guide the fine-tuning of stronger models.

**Strengths:**

WSPO is broadly applicable to Reinforcement Learning with Human Feedback (RLHF) tasks, and the benchmark results validate its effectiveness. Remarkably, even though a weaker model like Qwen2 performs poorly on its own (as shown in Table 1), the distribution changes induced by reference and DPO-tuned models provide meaningful guidance. This enables the stronger model to achieve significant performance improvements, demonstrating the method's potential to enhance model alignment despite the initial limitations of weaker models.

**Weaknesses:**

1. All experiments are conducted using the Qwen2 model family, and it would strengthen the study to include results from other open-source LLMs, (e.g., Llama3). Additionally, incorporating a Proximal Policy Optimization (PPO) baseline in Table 1 would provide a more comprehensive comparison.

2. While the proposed method and experiments are innovative and comprehensive from my perspective, the manuscript requires significant improvements in clarity and organization. Specifically, in Section 3.2, the process for obtaining $\pi_{r}^{weak}$ and $\pi_{ref}^{weak}$ should be explicitly detailed, as currently, **readers have to refer to the appendix** to understand their derivation through Supervised Fine-Tuning (SFT) and Direct Preference Optimization (DPO), and for the summarization task, via PPO.

3. Furthermore, the methods section needs clearer explanations of each notation upon the first introduction. The experimental section also lacks essential details and attempts to cover three sets of experiments simultaneously. To enhance readability and focus, it is recommended that Section 4.3 be emphasized with more comprehensive details, while at least one of the experiments in Sections 4.1 and 4.2 could be relocated to the appendix to streamline the presentation.

**Questions:**

1. The MMLU score (shown in Table 2) seems very low for QWen2 model, can you explain why it’s in the range of 40s?

2. Do you have experiment results for other model families?

3. You showed that alignment from an even weaker model like Qwen2-0.5B can be helpful. I am curious about what happens if the weaker model is not weak anymore (e.g. use 7B model’s SFT and DPO checkpoint to compute the ratio as if they are pi_{r}^{weak} and \pi_{ref}^{weak}).

---

> ### Author Response · Authors · 2024-11-20
> **Response to Reviewer ssa6**
>
> Thank you for your thoughtful and thorough review of our work.
>
> **[W2 & W3]** We have carefully revised the paper in accordance with your detailed suggestions and have uploaded the updated version, following the ICLR guidelines.
>
> > - We included a more detailed explanation of the notation and reorganized the structure of the paper for better clarity.
> > - Due to the significant computational demands of PPO training and challenges with training stability, we were unable to provide the results in time. Therefore, we retained both Part 4.1 and Part 4.2 as initially presented.
> >   - Exp 4.1 serves as a toy example to demonstrate how the model transitions from the base model to the aligned model can be achieved using WSPO in some cases.
> >   - Exp 4.2 highlights the training stability of the WSPO algorithm.
> >   - Exp 4.3 provides a comprehensive evaluation.
>
> **[Q1.]	Because of the configuration of llm-eval-harness repo.**
>
> > For the MMLU dataset, the llm-evaluation-harness offers various evaluation settings.  In our paper, we initially used the "continuation setting," where the model is not presented with answer choices for the questions.  However, we have since switched to the "default setting" and updated the corresponding results in our paper.
> >
> > |      | Qwen2-7B-base | Qwen2-7B-Instruct | Qwen2-7B-base + WSPO | Qwen2-7B-Instruct + WSPO |
> > | ---- | ------------- | ----------------- | -------------------- | ------------------------ |
> > | MMLU | 69.43         | 69.94             | 69.37                | 69.44                    |
>
> **[W1 & Q2.]	Experiments for Llama3 families.**
>
> > **[Experiment]** LLaMA 3.2 released lightweight 1B and 3B models on October 24, 2024. We conducted the following experiments using these models. We utilize 1B weak models to align 3B and 8B models, with the hyperparameters for each method specified in parentheses.
> >
> > | Arena-hard          | Llama3.2-1B-Instruct | Llama3.2-3B-Instruct | Llama3.1-8B -Instruct |
> > | ------------------- | -------------------- | -------------------- | --------------------- |
> > | Instruct            | 11.40                | 29.60                | 50.30                 |
> > | dpo ($\beta=0.1$)   | 11.70                | 31.70                | 48.40                 |
> > | wspo ($\gamma=0.5$) | ---                  | **31.80**            | **50.80**             |
> >
> > | Arena-hard          | Llama3.2-1B-Instruct                      | Llama3.2-3B-Instruct                                         | Llama3.1-8B -Instruct                                        |
> > | ------------------- | ----------------------------------------- | ------------------------------------------------------------ | ------------------------------------------------------------ |
> > | Instruct            | 11.40                                     | 29.60                                                        | 50.30                                                        |
> > | SimPO               | 14.60 ($\beta=2.5, \gamma /\beta = 0.55$) | 0.70 ($\beta=2.5, \gamma /\beta = 0.55$) ;26.50 ($\beta=10, \gamma /\beta = 0.30$) | 0.00 ($\beta=2.5, \gamma /\beta = 0.55$) ;3.50 ($\beta=10, \gamma /\beta = 0.30$) |
> > | wspo ($\gamma=0.5$) | ---                                       | **31.20**                                                    | **52.60**                                                    |
>
> **[Q3.]	It also has an alignment enhancement phenomenon.**
>
> > We use 7B model’s SFT and DPO checkpoint to compute the ratio as if they are $\pi_{r}^{weak}$ and $\pi_{ref}^{weak}$ and using this ratio as the label to align 7B model's SFT checkpoint, we derive the following results.
> >
> > | Arena-hard          | Qwen2-7B-Base |
> > | ------------------- | ------------- |
> > | SFT                 | 4.70          |
> > | DPO ($\beta=0.1$)   | 11.20         |
> > | WSPO ($\gamma=1.0$) | 10.90         |
> > | WSPO ($\gamma=0.5$) | **14.90**     |
> > | WSPO ($\gamma=0.1$) | **15.30**     |
> >
> > When ($\gamma = 1.0$), the alignment strength is nearly identical to the DPO-aligned model's. However, as the alignment strength is reduced relative to the original DPO-trained model, an enhancement in alignment is observed.
>
> **We sincerely look forward to engaging in further discussions.**

---

> ### Author Response · Authors · 2024-11-25
>
> Dear Reviewer ssa6,
>
> Thank you again for your time and efforts in reviewing our paper.
>
> As the discussion period draws close, we kindly remind you that two days remain for further comments or questions. We would appreciate the opportunity to address any additional concerns you may have before the discussion phase ends.
>
> Thank you very much.
>
> Best regards,
>
> Authors

---

> > ### Comment · Reviewer_ssa6 · 2024-11-25
> > **Official Comment by Reviewer ssa6**
> >
> > Thanks for addressing my question and providing additional experiments and analyses. I will increase my score to 8.

---

> > > ### Author Response · Authors · 2024-11-28
> > >
> > > We are happy to hear that our rebuttal addressed your questions well. Thank you very much for your timely response and for increasing the score.

---

### Official Review · Reviewer_LZuG · 2024-11-01

**Soundness:** 3
**Presentation:** 3
**Contribution:** 3
**Rating:** 8
**Confidence:** 4

**Summary:**

The paper introduces a novel method called Weak-to-Strong Preference Optimization (WSPO), which aims to align large language models (LLMs) with human preferences by leveraging the alignment signals of weaker models. The proposed method builds on the concept of weak-to-strong generalization, where a strong model can amplify the alignment capability learned from a weaker model. WSPO works by learning the distributional differences before and after the alignment of a weaker model, which enables a stronger model to effectively inherit and improve upon the alignment behavior of the weaker one. The authors demonstrate that WSPO improves the win rate and alignment quality of the Qwen2-7B model in several benchmarks, including Arena-Hard, AlpacaEval 2, and MT-bench.

**Strengths:**

Novelty: WSPO is a unique approach that effectively transfers alignment signals from weak to strong models, providing an alternative to reward model training. This is a valuable addition to existing alignment techniques such as RLHF and DPO.

Quality: The experimental results demonstrate that WSPO performs competitively with strong models trained using PPO, and in some cases even surpasses it in alignment capabilities.

Clarity: The paper provides a well-organized discussion of prior work, situating WSPO within the broader context of model alignment. The use of diagrams to illustrate alignment processes helps in understanding the proposed method.

Significance: WSPO has practical significance for reducing the computational cost of aligning strong models, as it eliminates the need for repeated reward model training.

**Weaknesses:**

Complexity of Theoretical Derivations: The theoretical sections, particularly those related to deriving the WSPO objective and gradient, are quite dense and might be challenging for readers without a strong background in reinforcement learning. Consider providing more intuitive explanations or additional examples.

**Questions:**

1. The method relies on aligning the weak model first and then transferring this alignment to a stronger model. Are there scenarios where aligning the strong model directly would be more beneficial, and how does WSPO compare in terms of resource efficiency in such cases?
2. Is WSPO applicable to non-language model tasks, such as reinforcement learning agents or vision-language tasks? It would be interesting to understand the generalizability of the approach beyond language models.

---

> ### Author Response · Authors · 2024-11-20
> **Response to Reviewer LZuG**
>
> Thank you for your thoughtful and thorough review of our work.
>
> **[For weakness.]** We have provided an intuitive explanation in the Method section and refined our description to make the theoretical derivations clearer and easier to understand. See the updated paper for details.
>
> **[Q1.]	Here are three scenarios:**
>
> > - **Abundant high-quality, preference-chosen dataset:**
> >   In this scenario, we can directly apply SFT to align the strong model. However, our method needs to incorporate additional weak models to aid the strong model's learning process.
> >
> > - **Abundant computational resources, robust reward model, and strong expertise in model training:**
> >   In this case, PPO enables the model to explore unseen distributions within the dataset, enhancing its robustness. Here, our method demonstrates greater efficiency.
> > - **Direct preference optimization:**
> >   Techniques like SimPO offer a lightweight optimization approach. However, SimPO relies on two ad-hoc hyperparameters and, also requires a **high-quality preference dataset**. In contrast, our method is more invariant to hyperparameters and has stringent requirements on the dataset.
>
> **[Q2.]	WSPO can apply to the probabilistic models**
>
> > - Current vision-language tasks, are commonly composed of two components, one is an auto-regressive language model, another is an image encoder to extract representations into the core LLM (e.g. Llama 3.2-Vision[1]).
> > - LMs can be viewed as agents with a large decision space (i.e., the vocabulary size). In contrast, RL agents often operate within a much smaller decision space. However, as long as these agents rely on probabilistic modeling, we believe our method can be effectively applied. Our approach focuses on learning the distributional differences before and after fine-tuning, making it suitable for such scenarios.
> >
> > **[Experiment]** We use the RLHF-v dataset (image-text pair preference dataset) to perform DPO and WSPO based on the vision-language models. We use 2B models to align 7B models. The evaluation results are shown as follows.
> >
> > | Arena-hard | Qwen2-2b-VL | Qwen2-7b-VL |
> > | ---------- | ----------- | ----------- |
> > | Sft        | 1.40        | 5.30        |
> > | DPO        | 1.50        | 4.90        |
> > | WSPO       | ---         | **5.80**    |
> >
> > MMHal-Bench is a dataset consisting of image-question pairs, designed to assess hallucinations and response informativeness. Below are the evaluation results for Qwen2-7b-VL.
> >
> > | MMhal-Bench | Informativeness ($\Uparrow $, full score: 6) | Hullucination rate (($\Downarrow $, full score: 1)) |
> > | ----------- | -------------------------------------------- | --------------------------------------------------- |
> > | Sft         | 3.91                                         | 0.23                                                |
> > | DPO         | 3.80                                         | 0.27                                                |
> > | WSPO        | **4.02**                                     | **0.22**                                            |
> >
> > **Reference:**
> >
> > [1]. https://huggingface.co/meta-llama/Llama-3.2-11B-Vision
>
> **We sincerely look forward to engaging in further discussions.**

---

> ### Author Response · Authors · 2024-11-25
>
> Dear Reviewer LZuG,
>
> Thank you again for your time and efforts in reviewing our paper.
>
> As the discussion period draws close, we kindly remind you that two days remain for further comments or questions. We would appreciate the opportunity to address any additional concerns you may have before the discussion phase ends.
>
> Thank you very much.
>
> Best regards,
>
> Authors

---

### Official Review · Reviewer_oPbS · 2024-11-01

**Soundness:** 3
**Presentation:** 4
**Contribution:** 3
**Rating:** 8
**Confidence:** 4

**Summary:**

The paper presents a novel approach called Weak-to-Strong Preference Optimization (WSPO), designed to enhance model alignment by leveraging alignment signals from weaker models to improve stronger ones. Inspired by weak-to-strong generalization, WSPO transfers the alignment abilities of a weaker model to a stronger one by optimizing the differences in distributions before and after weak model alignment. WSPO demonstrated competitive results across various benchmarks, showing a significant performance improvement over existing approaches, such as Direct Preference Optimization (DPO).

**Strengths:**

1. **Innovation in Model Alignment**: WSPO introduces a unique method of transferring alignment signals from weak to strong models, showcasing effective model alignment without the need for a dedicated reward model.
2. **Experimental Validation**: The paper provides thorough experimental evidence across several tasks, including summarization, single-turn dialogue, and complex benchmarks like MT-Bench and AlpacaEval 2. The results demonstrate WSPO's strong performance relative to baselines like DPO.
3. **Efficiency**: WSPO presents a more streamlined alignment process, avoiding the costly RLHF approach. This approach may offer practical advantages for resource-constrained settings, as the alignment signal is derived directly from weaker models.
4. **Comprehensive Analysis**: The authors conduct various analyses, including the impact of model size and the role of key hyperparameters, providing insights into the approach's scalability and stability.

**Weaknesses:**

1. **Limited Cross-Model Analysis**: While the paper demonstrates the impact of weak-to-strong transfer within a specific model family, it lacks analysis of WSPO’s effectiveness across diverse architectures.
2. **Lack of Theoretical Explanation**: Although the approach works empirically, the authors do not provide a detailed theoretical rationale for why transferring alignment from weak to strong models amplifies alignment effectiveness.
3. **Reliance on Specific Datasets**: The experimental results heavily rely on specific datasets (UltraChat, Anthropic HH), potentially limiting the generalizability of WSPO to other alignment contexts. Broader dataset testing could validate the approach further.
4. **Parameter Sensitivity**: The method’s effectiveness appears sensitive to hyperparameter tuning (e.g., γ in WSPO), which might complicate deployment across different model types and alignment scenarios.

**Questions:**

1. Have you tested WSPO across different model architectures or families beyond Qwen? If so, could you provide insights into its effectiveness across various architectures?

2. Could you explain why the alignment capability amplifies when transferred from a weak model to a strong model? Are there theoretical or empirical insights that could clarify this phenomenon?

3. Given WSPO's reliance on certain hyperparameters (e.g., γ), do you have any practical guidelines for setting these across different model sizes or tasks? Could automatic tuning methods be integrated to simplify deployment?

4. Could you provide further details on WSPO’s computational efficiency, particularly in comparison to RLHF in real-world applications? How significant are the resource savings, and are there trade-offs in alignment quality for complex, large-scale tasks?

5. How critical is the quality of the initial alignment in the weak model to WSPO's success? Would a weak model with suboptimal alignment still provide useful signals for the strong model?

---

> ### Author Response · Authors · 2024-11-20
> **Response to Reviewer oPbS [1/2]**
>
> Thank you for your thoughtful and thorough review of our work.
>
> **[W3. ]	Our method is to learn the predicted distribution difference before and after model alignment, rather than relying on a specific dataset.**
>
> > However, we should acknowledge the importance of weak model alignment. After all, great models are built on great data.
> >
> > **[Experiment]** To illustrate this, we use the **rejected** subset of the preference dataset, which may include **toxic content**. This dataset performs WSPO training to capture the predicted distribution difference. **The results show that our method does not depend on the specific datasets.**
> >
> > | Arena-hard          | Qwen2-1.5B - Instruct | Qwen2-7B-Instruct |
> > | ------------------- | --------------------- | ----------------- |
> > | SFT                 | 1.30                  | 34.40             |
> > | DPO  ($\beta=0.1$)  | 3.40                  | 34.00             |
> > | WSPO ($\gamma=0.5$) | --                    | **40.30**         |
>
> **[W1 & Q1.]	Experiments for Llama3 families.**
>
> > **[Experiment]** LLaMA 3.2 released lightweight 1B and 3B models on October 24, 2024. We utilize 1B weak models to align 3B and 8B models, with the hyperparameters for each method specified in parentheses.
> >
> > | Arena-hard          | Llama3.2-1B-Instruct | Llama3.2-3B-Instruct | Llama3.1-8B -Instruct |
> > | ------------------- | -------------------- | -------------------- | --------------------- |
> > | Instruct            | 11.40                | 29.60                | 50.30                 |
> > | dpo ($\beta=0.1$)   | 11.70                | 31.70                | 48.40                 |
> > | wspo ($\gamma=0.5$) | ---                  | **31.80**            | **50.80**             |
> >
> > | Arena-hard          | Llama3.2-1B-Instruct                      | Llama3.2-3B-Instruct                                         | Llama3.1-8B -Instruct                                        |
> > | ------------------- | ----------------------------------------- | ------------------------------------------------------------ | ------------------------------------------------------------ |
> > | Instruct            | 11.40                                     | 29.60                                                        | 50.30                                                        |
> > | SimPO               | 14.60 ($\beta=2.5, \gamma /\beta = 0.55$) | 0.70 ($\beta=2.5, \gamma /\beta = 0.55$) ;26.50 ($\beta=10, \gamma /\beta = 0.30$) | 0.00 ($\beta=2.5, \gamma /\beta = 0.55$) ;3.50 ($\beta=10, \gamma /\beta = 0.30$) |
> > | wspo ($\gamma=0.5$) | ---                                       | **31.20**                                                    | **52.60**                                                    |
> >
> > Section 3 of the paper [1] utilizes DPO as a distillation objective to transfer the transformer-based alignment capability into Mamba. Similarly, WSPO can also transfer alignment capabilities between models with different architectures.
> >
> > **Reference**
> >
> > [1] The Mamba in the Llama: Distilling and Accelerating Hybrid Models, https://arxiv.org/abs/2408.15237
>
> **[W2 & Q2. ]	Some Empirical insights into alignment enhancement**
>
> > - The weak model's parameter size might constrain weak model alignment, but transferring this alignment to stronger models may yield significant benefits.
> > - Our method avoids direct training on the preference dataset, thereby mitigating risks like **overfitting** and **reward hacking**.
> > - Model alignment methods such as DPO, SimPO, and others may lead to suboptimal alignment performance. In contrast, our approach can **control alignment strength** using the hyperparameter $\gamma$, which adjusts the alignment strength ratio between the strong and weak models.
>
> **[W4 & Q3.]	Guidelines for hyperparameter**
>
> > - **[Practical Guidelines]** In the paper, we derive that $\gamma = \lambda / \beta$, which represents the ratio of alignment regularization between the strong model and the weak model. Using weak models to align stronger models, typical choices like $\gamma = 0.1$ or $\gamma = 0.5$ work well. As mentioned in our paper, when $\gamma > 1$, the aligned strong models tend to favor the reference distribution. When $\gamma = 1$, the aligned strong models mirror the weakly aligned models. Notably, when $\gamma < 1$, alignment performance is often significantly improved.
> > - **[Automatic Tuning Methods]** One automated approach we can use is the adaptive-KL algorithm. This involves maximizing the reward term, $\beta \log \frac{\pi_r^{\text{weak}}}{\pi_{ref}^{\text{weak}}}$, and dynamically adjusting $\lambda$ .For example,  $\lambda \leftarrow \lambda \cdot \alpha  \text { if } KL >KL_{\text {target }}$ Or $\lambda \leftarrow \lambda / \alpha \text { if } KL<KL_{\text {target }} $ $\alpha $ is the adjustment factor.
>
> **We sincerely look forward to engaging in further discussions.**

---

> ### Author Response · Authors · 2024-11-20
> **Response to Reviewer oPbS [2/2]**
>
> **[Q4.]	Computational efficiency compare to RLHF**
>
> > 1. **Training a Reward Model:**
> >
> >    - For **PPO** in RLHF, we train a 1.5B reward model by adding a layer to the base LM to predict reward values.
> >    - For **WSPO**, we also train a 1.5B reward model using approaches like DPO, SimPO, and other algorithms.
> >
> >    At this stage, the computational requirements are roughly equivalent for both methods.
> >
> > 2. **Using the Reward Model for Training:**
> >
> >    - Once the reward model is trained, we use it to train both PPO and WSPO.
> >    - Performance benchmarking is conducted on a single node with 8x H100 GPUs under the following configuration:
> >      - Batch size: 32, Sequence length: 4K, Training steps: 5724
> >
> >    The measured training times are as follows:
> >
> >    | Model Size | PPO          | WSPO           |
> >    | ---------- | ------------ | -------------- |
> >    | 7B         | 95–120 hours | **54 minutes** |
> >
> > **[Trade-off Analysis]** Currently, we can use WSPO optimization as a precursor to PPO, as it may significantly accelerate PPO training. While PPO is more computationally intensive, it remains the most robust approach, enabling the exploration of optimizations beyond the dataset's distribution. Thus, PPO can serve as a valuable method to further enhance model performance.
>
> **[Q5.]	Weak model alignment is important**
>
> > - **Weak model alignment is crucial.** If a weakly aligned model perfectly matches the reference model, it will cease to perform gradient optimization effectively.
> > - **Suboptimal models can still offer valuable insights.** Weak model alignment might be constrained by the model's parameter size, but transferring this alignment to stronger models can yield significant benefits.
> >
> > **[Experiment]** We use 0.5B models to align 7B models, while the alignment achieved by the 0.5B model is limited.
> >
> > | Arena-hard             | Qwen2-0.5B - Instruct | Qwen2-7B-Instruct |
> > | ---------------------- | --------------------- | ----------------- |
> > | SFT                    | 0.90                  | 34.40             |
> > | DPO  ($\beta = 0.1$)   | 0.90                  | 34.00             |
> > | WSPO  ($\gamma = 0.1$) | --                    | **40.60**         |
>
> **We sincerely look forward to engaging in further discussions.**

---

> ### Author Response · Authors · 2024-11-25
>
> Dear Reviewer oPbS,
>
> Thank you again for your time and efforts in reviewing our paper.
>
> As the discussion period draws close, we kindly remind you that two days remain for further comments or questions. We would appreciate the opportunity to address any additional concerns you may have before the discussion phase ends.
>
> Thank you very much.
>
> Best regards,
>
> Authors

---

> > ### Comment · Reviewer_oPbS · 2024-12-02
> > **Reponse to rebuttal**
> >
> > Thank you for your responses and addressing my concerns. The new experiments validate the approach further, so I have decided to raise my score.

---

> > > ### Author Response · Authors · 2024-12-03
> > >
> > > We are happy to hear that our rebuttal addressed your questions well. Thank you very much for your timely response and for increasing the score.

---

### Official Review · Reviewer_4UGn · 2024-11-05

**Soundness:** 3
**Presentation:** 3
**Contribution:** 3
**Rating:** 5
**Confidence:** 5

**Summary:**

The paper presents Weak-to-Strong Preference Optimization (WSPO), a pioneering approach for aligning language models (LMs) with human preferences by leveraging the alignment capabilities of weaker models to enhance stronger ones. Through learning distributional differences before and after the alignment of the weaker model, WSPO effectively transfers and amplifies alignment behaviors, demonstrating improved performance on key benchmarks such as Arena-Hard, AlpacaEval 2, and MT-bench.

**Strengths:**

1. Innovative Transfer of Alignment Capability: WSPO innovatively transfers alignment capabilities from a weaker model to a stronger one, which is a novel approach in the field of LM alignment.
2. Performance Improvement: The paper shows significant improvements in win rates on challenging benchmarks, indicating the effectiveness of WSPO in aligning stronger models.
3. Empirical Evidence: The paper provides extensive experimental results, demonstrating WSPO's effectiveness across various tasks and datasets, including summarization, dialogue, and reasoning tasks.
4. Generalization Beyond Weak Labels: WSPO not only imitates weak model behavior but also generalizes beyond it, as shown by the strong model's improved performance on benchmarks.

**Weaknesses:**

1. Dependence on Quality of Weak Model Alignment: The success of WSPO is heavily dependent on the quality of alignment in the weak model, which might not always be optimal.
2. Hyperparameter Sensitivity: The performance of WSPO seems to be sensitive to the choice of hyperparameters like γ, which might require careful tuning for different models and tasks.
3. Efficiency Considerations: During WSPO training, the algorithm requires the loading of parameters from four models, two of which are weak models with limited parameters. The paper omits a comparison with previously established efficient methods like SimPO, which could potentially deliver better performance and higher training efficiency.

**Questions:**

see weakness.

---

> ### Author Response · Authors · 2024-11-20
> **Response to Reviewer 4UGn**
>
> Thank you for your thoughtful and thorough review of our work.
>
> **[W1.]	Suboptimal models can still offer valuable insights.**
>
> > - If a weakly aligned model perfectly matches the reference model, it will no longer perform gradient optimization.
> > - However, aligning a weak model, as opposed to a strong one, provides greater flexibility in discovering a relatively optimal aligned model.
> > - Weak model alignment might be constrained by the model's parameter size, but transferring this alignment to stronger models can yield significant benefits.
> > - Our method introduces a way to **adjust the alignment strength** via the hyperparameter $\gamma$, which represents the ratio of regularization strength between the strong and weak models.
> >
> > **[Experiment]** We use 0.5B models to align 7B models, while the alignment achieved by the 0.5B model is limited.
> >
> > | Arena-hard             | Qwen2-0.5B - Instruct | Qwen2-7B-Instruct |
> > | ---------------------- | --------------------- | ----------------- |
> > | SFT                    | 0.90                  | 34.40             |
> > | DPO  ($\beta = 0.1$)   | 0.90                  | 34.00             |
> > | WSPO  ($\gamma = 0.1$) | --                    | **40.60**         |
>
> **[W2.]	 Our hyperparameter is easy to understand.**
>
> > - Our loss function contains just one hyperparameter. In Section 5.2 of the Analysis paper, we visualize how different hyperparameter values influence the experimental results. Typically, values like **0.1 or 0.5** work well (e.g., for Qwen, Llama, etc.).
> > - This hyperparameter is easy to understand. It helps maintain alignment with the reference distribution while balancing the regularization strength between strong and weak models.
> >
> > **[Experiment]**  We selected the Llama3-8B-Instruct model as our weak reference model and the pre-aligned Simpo-aligned model from the SimPO project [1] as the weakly aligned model. **By simply setting the hyperparameter $\gamma = 1$,** we performed WSPO based on the Llama3-8B-Instruct model. The results are presented below, and, surprisingly, we also observed an alignment enhancement effect. **This futther proves that in [W1],** our method has the effect of adjusting the alignment strength.
> >
> > | Arena-Hard                                | LLama3-8B-Instruct |
> > | ----------------------------------------- | ------------------ |
> > | SFT                                       | 38.90              |
> > | SimPO ($\beta=2.5, \gamma /\beta = 0.55$) | 52.20              |
> > | WSPO ($\gamma = 1.0$)                     | **53.80**          |
>
> **[W3. ]	Trade-off between efficiency and parameter-tuned tasks.**
>
> > - While we recognize that SimPO is lightweight, it requires tuning two hyperparameters, for which clear guidelines are lacking. As noted in the *Hyperparameter Tuning* section of their project page [1], this process can be challenging.
> > - One of the key contributions of our work is to prove that we can utilize the predicted distributions before and after model alignment as labels to guide the alignment process. As a result, we do not focus on comparing various advanced alignment algorithms.
> >
> > **[Experiment]** We utilize 1B weak models to align 3B and 8B models, with the hyperparameters for each method specified in parentheses. SimPO hyperparameters are selected from their project page[1]. As shown in the following table, **if slightly more resources are available, a method requiring less human intervention may be preferable.**
> >
> > | Arena-hard          | Llama3.2-1B-Instruct                      | Llama3.2-3B-Instruct                                         | Llama3.1-8B -Instruct                                        |
> > | ------------------- | ----------------------------------------- | ------------------------------------------------------------ | ------------------------------------------------------------ |
> > | Instruct            | 11.40                                     | 29.60                                                        | 50.30                                                        |
> > | SimPO               | 14.60 ($\beta=2.5, \gamma /\beta = 0.55$) | 0.70 ($\beta=2.5, \gamma /\beta = 0.55$) ;26.50 ($\beta=10, \gamma /\beta = 0.30$) | 0.00 ($\beta=2.5, \gamma /\beta = 0.55$) ;3.50 ($\beta=10, \gamma /\beta = 0.30$) |
> > | wspo ($\gamma=0.5$) | ---                                       | **31.20**                                                    | **52.60**                                                    |
> >
> > **Reference:**
> >
> > [1] https://github.com/princeton-nlp/SimPO
>
> **We sincerely look forward to engaging in further discussions.**

---

> ### Author Response · Authors · 2024-11-25
>
> Dear Reviewer 4UGn,
>
> Thank you again for your time and efforts in reviewing our paper.
>
> As the discussion period draws close, we kindly remind you that two days remain for further comments or questions. We would appreciate the opportunity to address any additional concerns you may have before the discussion phase ends.
>
> Thank you very much.
>
> Best regards,
>
> Authors

---

### Author Response · Authors · 2024-11-28
**Paper Revision Summary**

**We greatly appreciate the reviewers' thoughtful, high-quality review of our work.**

**All reviewers** found our work **innovative**  and **novel**. Reviewer `4UGn` described it as "**pioneering,**" while Reviewers `oPbS` and `LZuG` described it as "**unique.**"

As the revision deadline approaches, we present the following summary of the key updates.

------

**Revisions to Address Q&W:**

1. **[Content Highlighted in Red]** Improved clarity of notation and simplified the method description (`addressing Reviewers LZuG and ssa6`).
2. **[Experiment 4.3.2]** Added empirical insights into alignment enhancement (`addressing Reviewer oPbS`).
3. **[Analysis 5.1]** Demonstrated that suboptimal weak model can still provide alignment signal (`addressing Reviewers 4UGn, oPbS`).
4. **[Analysis 5.2, Appendix D]** Provided guidelines for hyperparameter tuning (`addressing Reviewers 4UGn, oPbS`).
5. **[Appendix B]** Included experiments with Llama families, further showing the effectiveness of WSPO (`addressing Reviewers oPbS, ssa6`).
6. **[Appendix C]** Analyzed the impact of the dataset, demonstrating that WSPO focuses on learning distributional differences rather than relying on a specific dataset (`addressing Reviewer oPbS`).
7. **[Appendix D]** Evaluated equivalent models as weak models, showing WSPO's role in adjusting alignment strength (`addressing Reviewer ssa6`).
8. **[Appendix E]** Extended WSPO to vision-language tasks, showing its applicability beyond language models (`addressing Reviewer LZuG`).
9. **[Appendix F]** Added an efficiency comparison with SimPO and RLHF (`addressing Reviewers 4UGn, oPbS`).

We sincerely hope these **comprehensive updates** meet the reviewers' expectations and look forward to receiving positive feedback in the remaining discussion period.

---

### Meta-Review · Area_Chair_9zS3 · 2024-12-20

**Metareview:**

This paper presents a new approach to alignment where the larger model's reward is regressed against the smaller model's reward, where both the reward models are parameterized as a ratio of the policy and the sft model (i.e., as in DPO). The authors find that this approach can result in stronger aligned models than in regular DPO.

On the positive side, this paper presents an interesting approach to alignment that is coupled with some surprising empirical results. (I was genuinely surprised to see this approach outperform a DPO-aligned stronger model.) On the negative side, this approach introduces additional overhead (since the smaller models need to be loaded), and is missing obvious baselines (e.g., PPO with the smaller model as the reward). I also think there is a substantial ensembling effect happening with this model, which is not discussed by the authors (e.g., I bet an ensemble of the strong-DPO and weak-DPO model will do quite well).

But given the strong(ish) empirical results, I think this paper should be accepted.

**Additional Comments On Reviewer Discussion:**

Most reviewers were on the fence after the initial review, but became more enthusiastic after the authors conducted additional experiments during the rebuttal period. I also appreciated the authors' extensive experiments to address reviewer concerns.

---

### Decision · Program_Chairs · 2025-01-22

Accept (Spotlight)